# Conservation Agriculture and Soil Organic Carbon: Principles, Processes, Practices and Policy Options

Rosa Francaviglia [1],*, María Almagro [2] and José Luis Vicente-Vicente [3]

1   CREA, Research Centre for Agriculture and Environment, 00184 Rome, Italy
2   IFAPA, Andalusian Institute of Agricultural and Fisheries Research and Training, Camino de Purchil s/n, 18004 Granada, Spain
3   ZALF, Leibniz Centre for Agricultural Landscape Research, 15374 Müncheberg, Germany
*   Correspondence: r.francaviglia@gmail.com

**Abstract:** Intensive agriculture causes land degradation and other environmental problems, such as pollution, soil erosion, fertility loss, biodiversity decline, and greenhouse gas (GHG) emissions, which exacerbate climate change. Sustainable agricultural practices, such as reduced tillage, growing cover crops, and implementing crop residue retention measures, have been proposed as cost-effective solutions that can address land degradation, food security, and climate change mitigation and adaptation by enhancing soil organic carbon (SOC) sequestration in soils and its associated co-benefits. In this regard, extensive research has demonstrated that conservation agriculture (CA) improves soil physical, chemical, and biological properties that are crucial for maintaining soil health and increasing agroecosystem resilience to global change. However, despite the research that has been undertaken to implement the three principles of CA (minimum mechanical soil disturbance, permanent soil organic cover with crop residues and/or cover crops, and crop diversification) worldwide, there are still many technical and socio-economic barriers that restrict their adoption. In this review, we gather current knowledge on the potential agronomic, environmental, and socio-economic benefits and drawbacks of implementing CA principles and present the current agro-environmental policy frameworks. Research needs are identified, and more stringent policy measures are urgently encouraged to achieve climate change mitigation targets.

**Keywords:** reduced tillage; permanent soil cover; crop diversification; soil and water conservation; ecosystem services; carbon sequestration; climate change mitigation and adaptation; adoption barriers; economic incentives; agro-environmental policies

## 1. Background and Rationale

The concept of conservation agriculture (CA) was born in the 1930s when Edward Faulkner first questioned the utility of ploughing in a manuscript called *Ploughman's Folly*, and it gained popularity during the 1960s in the mid-western United States as a means of preventing soil degradation after the Dust Bowl ecological disaster that occurred in the 1930s. Since then, research on adapting CA practices to cropping systems has been undertaken worldwide. In addition to reducing tillage intensity, CA also implies the application of organic amendments, such as manure, compost, and by-products from agro-industry [1], and the improvement of N management if mineral fertilizers are adopted to decrease $N_2O$ emissions [2].

The exploitation of agricultural soils based on crop monocultures and deep tillage with inversion of the layers has resulted in progressive soil structure degradation and compaction and reductions in soil organic matter content. These detrimental developments have triggered negative cascade effects on the soil biota and fertility, increasing soil water and wind erosion and $CO_2$ emissions [3,4]. Among alternative management systems to conventional agriculture that aim at the sustainability of crop systems, CA represents one of the most advanced models.

## 2. Adoption of Conservation Agriculture (CA)

CA is defined by the Food and Agriculture Organization [5,6] as "a sustainable agricultural production system for the protection of water and agricultural soil that integrates agronomic, environmental and economic aspects". CA is based on three principles (Figure 1): minimum mechanical soil disturbance through conservation tillage (i.e., no tillage, minimum tillage), permanent soil organic cover with crop residues and/or cover crops, and crop diversification through rotations and associations involving at least three different crops (including a legume crop). The benefits of CA are shown in Table 1.

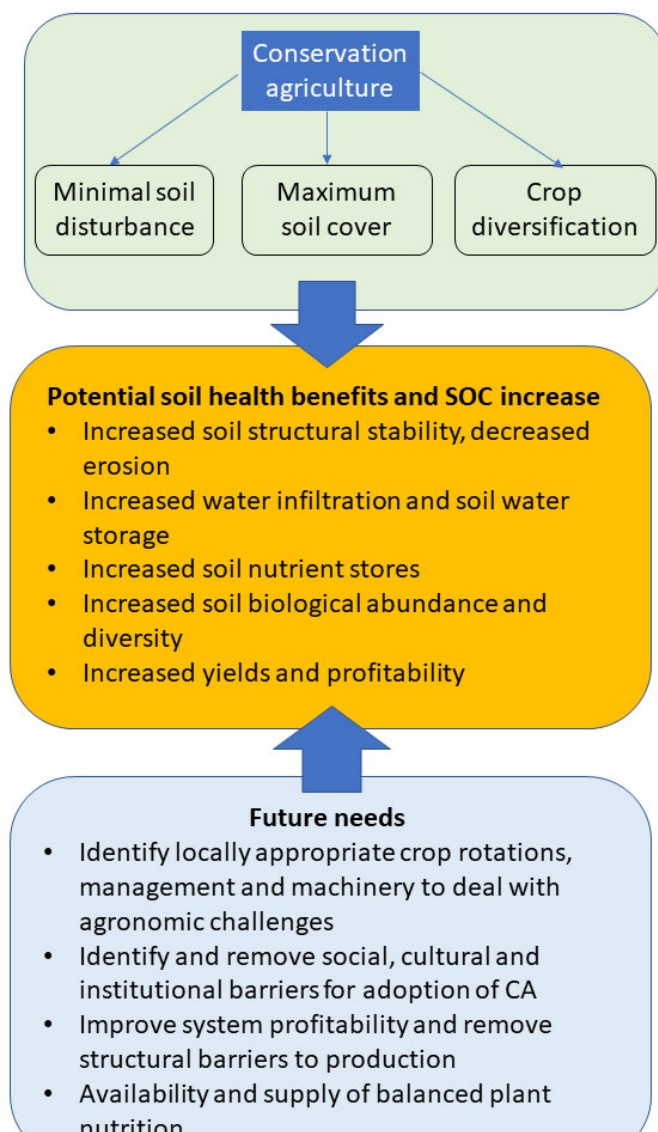

**Figure 1.** Principles of conservation agriculture, benefits of increasing SOC, and future needs. Modified from [7].

Kassam et al. [8] analysed the spread of the adoption of CA in 2015–2016 in different countries based on data available from government statistics, no-till farmer organizations, ministries of agriculture, non-governmental organizations, and research and development organizations.

The highest cropland areas were in South and North America (Table 2), with 69.9 and 63.2 M ha of cropland areas employed for CA, representing 38.7 and 35.0% of the total cropland employed for CA, respectively. However, CA represented 63.2% of the

cropland area in South America and 28.1% in North America. The corresponding values for Australia/New Zealand and Asia were 22.7 and 13.9 M ha (12.6 and 7.7% of total cropland), representing 45.4 and 4.1% of croplands in the respective regions. Cropland areas employed for CA decreased in the order Russia/Ukraine > Europe > Africa from 5.7 to 1.5 M ha; i.e., from 3.6 to 1.1% of these regions' total cropland areas, respectively. Globally, the total cropland area employed for CA was 180.4 M ha, equivalent to 12.5% of total cropland.

**Table 1.** Benefits of conservation agriculture [9].

| Target | Soil Cover | Minimal or No Soil Disturbance | Legumes in the Rotation | Crop Diversification |
|---|---|---|---|---|
| Simulate "forest floor" conditions | X | X | | |
| Reduce evaporative loss of moisture from soil surface | X | | | |
| Reduce evaporative loss from upper soil layers | X | X | | |
| Minimize oxidation of SOM and $CO_2$ loss | | X | | |
| Minimize compaction due to intense rainfall and the passage of machinery | X | X | | |
| Minimize temperature fluctuations at the soil surface | X | | | |
| Maintain supply of OM as substrate for soil biota | X | | | |
| Increase and maintain nitrogen levels in the root zone | X | X | X | X |
| Increase CEC of the root zone | X | X | X | X |
| Maximize rain infiltration and minimize runoff | X | X | | |
| Minimize soil loss in runoff | X | X | | |
| Maintain natural layering of soil horizons through actions of soil biota | X | X | | |
| Minimize weeds | X | X | | X |
| Increase rate of biomass production | X | X | X | X |
| Speed up recuperation of soil porosity by soil biota | X | X | X | X |
| Reduce labour input | | X | | |
| Reduce fuel-energy input | | X | | |
| Recycle nutrients | X | X | X | X |
| Reduce pests and diseases | | | | X |
| Rebuild damaged soil conditions and dynamics | X | X | X | X |

**Table 2.** Cropland areas employed for CA by region in 2015–16, CA area as percentage of global total cropland, and CA area as percentage of cropland of each region [8].

| Region | CA Cropland Area (M ha) | Total Cropland CA Area (%) | CA Area Cropland in the Region (%) |
|---|---|---|---|
| South America | 69.90 | 38.7 | 63.2 |
| North America | 63.18 | 35.0 | 28.1 |
| Australia/New Zealand | 22.67 | 12.6 | 45.5 |
| Asia | 13.93 | 7.7 | 4.1 |
| Russia/Ukraine | 5.70 | 3.2 | 3.6 |
| Europe | 3.56 | 2.0 | 5.0 |
| Africa | 1.51 | 0.8 | 1.1 |
| Global Total | 180.44 | 100 | 12.5 |

## 3. Principles: Conservation Tillage, Permanent Plant Cover, and Crop Diversification

### 3.1. Conservation Tillage (CT)

Tillage is needed for different agricultural processes (e.g., seedbed preparation, weed control, crop residue management, improving soil aeration and avoiding soil compaction, optimizing soil temperature and moisture regimes). However, as a consequence, soil physical and chemical properties (structure, bulk density, pore size distribution, and fertility condition) are also altered, ultimately leading to good or poor crop performance [10]. Appropriate tillage practices, such as CT, aim to avoid soil degradation without compromising crop yields and while maintaining agroecosystem stability [11].

CT, as defined by the Conservation Tillage Information Center (CTIC, West Lafayette, Indiana, USA), excludes those tillage operations that invert the soil and bury crop residues. It consists of reducing the ploughing depth occasionally or continuously, applying shallower tillage with other implements, and/or reducing the intensity of seedbed preparation. Thus, it minimizes soil disturbance and reduces losses in soil and water, for which at least

30% of the soil surface must be covered by crop residues. Therefore, CT is a general term that includes specific operations, such as no-tillage, minimum tillage, reduced tillage, and mulch tillage practices [12–14]. Interest in CT systems increased globally after the 1930s following the Dust Bowl events, as they were seen as a way to halt soil erosion and promote water conservation [15]. However, extensive research has further demonstrated the multiple environmental benefits of adopting CT, such as enhancement of soil organic carbon (SOC) content, maintenance of agricultural productivity, and savings in the costs—in terms of time, fuel, and machinery—of seedbed preparation [13,14]. Moreover, it has been demonstrated that leaving crop residues on the soil surface also reduces evapotranspiration, improves infiltration, and suppresses weed growth [12,16]. According to the CTIC, there are five types of CT systems.

(1)　No tillage (NT)

The NT system is a specialized type of CT consisting of a one-pass planting and fertilizer operation in which the soil and the surface residues are minimally disturbed [17]. NT systems eliminate all pre-planting mechanical seedbed preparation except for the opening of a narrow (2–3 cm wide) strip or small hole in the ground for seed placement that ensures adequate seed–soil contact [11]. Retaining crop residues and leaving them on the soil surface is pivotal for soil and water conservation. Weed control can be managed using herbicides, a brush cutter, or biological control methods, such as crop rotation, intercropping, or vegetation strips. However, the use of herbicides may have detrimental effects on the soil system and its functions; thus, they should be applied with caution. Indeed, the new European agro-environmental policy framework discourages the use of herbicides; thus, use of mechanical or biological control methods should be boosted. Among the potential benefits of NT compared to other tillage systems are that it is more effective in controlling soil erosion, it improves soil water storage capacity, and it results in lower energy costs per unit of production and higher grain yields, especially in low-slope areas. However, as already stated, major disadvantages of NT are the heavy use of herbicides for weed control and the risk of soil compaction and nutrient stratification [18,19] in intensive agricultural systems (e.g., low residue input, machine traffic).

(2)　Mulch tillage

Mulch tillage is based on the principles of causing the least disturbance to the soil and leaving the maximum percentage of crop residue on the soil surface. For this purpose, in addition to in situ crop residues, the use of live mulch derived from cover crop residues is becoming a common practice. This practice can be adopted in herbaceous and woody crop systems by either allowing spontaneous plant cover to become established or by growing cover crops in the fallow period (in the case of herbaceous crop systems) or in the inter-tree rows (in the case of woody crop systems). Regardless of the type of plant cover used, this practice consists of maintaining plant cover that can protect the soil for as long as possible without causing the problem of competition for water and nutrients with the main crop. To do so, in accordance with the crop type and climate conditions, the spontaneous or seeded plant covers are mowed before the water-limiting period starts, and their residues are left on the soil surface as mulch.

(3)　Strip or zonal tillage

Strip tillage is a practice in which soil disturbance is limited to the crop rows while the rest of the soil is left undisturbed [20]. This tillage practice emerged as an alternative soil management practice in attempts to solve and mitigate the problems derived from conventional tillage or direct seeding methods [21]. The seedbed is divided into a seedling zone (5–10 cm wide), which is mechanically tilled to optimize the soil and micro-climate environment for germination and establishment of seedlings, and an inter-row zone, which is left undisturbed and protected by mulch or managed using chiselling to improve water infiltration and root development [22]. Today, strip tillage can benefit from the use of global positioning system (GPS) guidance equipment [23].

(4)    Ridge till

Ridge tillage consists of leaving the soil undisturbed before planting and then tilling about one third of the soil surface when planting with sweeps or row cleaners. Crops are planted in rows on cultivated ridges, while weeds are controlled with herbicides. This tillage practice gained popularity as a conservation agriculture practice for maize and soybean production in the USA [17].

(5)    Reduced and minimum tillage or occasional tillage

Reduced tillage (RT) is a soil management practice that consists of reducing the total number of tillage passes per year needed before seed planting (in both annual and perennial crops) or for soil aeration and decompaction (particularly in perennial crops). RT is also called minimum tillage and shallow tillage since, in some cases, it refers to reducing the depth at which the soil is tilled and/or using a cultivator or chisel plough to avoid soil inversion. Occasional tillage refers to the practice of one-time tillage, where tillage is conducted once every 5 or 10 years—depending on the soil, climate, and crop type—in an otherwise continuous NT system. This tillage practice is generally applied to mitigate the potential negative effects that tillage cessation may cause in some cases, such as soil compaction and nutrient stratification, particularly in rainfed perennial cropping systems [24,25].

### 3.1.1. Context of Application

CT is presently applied worldwide under a wide variety of climate conditions and with a wide variety of soil types and crops. However, the potential benefits and drawbacks of CT vary with climate (dry vs. moist), soil type (clayey vs. sandy), crop type (arable vs. perennial), and management (rainfed vs. irrigated); therefore, CT must be locally adapted or combined with other practices to become more cost-effective. This practice has been widely adopted in humid, sub-humid, and tropical regions, particularly for arable crops. However, the adoption of CT can be a challenge in dry regions because of (1) low biomass production and (2) the fact that crop residues are needed as fuel or animal feed. Nevertheless, and despite the fact that CT has been proven to have several environmental benefits, there are still some limitations and barriers to overcome, as discussed in the following sections.

### 3.1.2. Potential for SOC Sequestration

The enhancement of SOC content when shifting from conventional (intensive) tillage to CT has been demonstrated worldwide [2,26]. However, variations in SOC sequestration rates can be found among studies depending on the climate conditions, soil characteristics, initial SOC levels, crop type (arable vs. woody cropping systems), management (rainfed vs. irrigated), and the duration of the experiments. Results from various meta-analyses and modelling studies indicate SOC sequestration rates ranging from 0.27 to 1.1 $t\,ha^{-1}\,yr^{-1}$ when CT is adopted in Mediterranean woody cropping systems [1,27,28]. Under Mediterranean conditions, average values about five times higher were reported for woody compared to arable crops for SOC sequestration rates [29]. Under tropical conditions, SOC sequestration rates oscillated between 0.12 and 1.56 $t\,SOC\,ha^{-1}\,yr^{-1}$ depending on the crop type and climate regime [2,26]. As would be expected, higher SOC sequestration rates were estimated for moist compared to dry conditions regardless of the crop type [26]. However, SOC sequestration rates were generally higher for arable compared to woody crops under tropical conditions. Under boreal conditions, a local study estimated SOC sequestration rates of between 0.28 and 0.39 $t\,ha^{-1}\,yr^{-1}$ across different soil types [30].

### 3.1.3. Co-Benefits

The enhancement of SOC content when shifting from conventional (intensive) tillage to CT has multiple beneficial effects, as has been demonstrated worldwide [2,27]. Several studies have demonstrated that CT improves soil physical, chemical, and biological properties crucial for maintaining soil condition and health. Indeed, conservation agriculture is indicated by the Intergovernmental Panel on Climate Change (IPCC) as one of the frameworks aimed at addressing land degradation, food security, and greenhouse gas (GHG) emissions [31]. For example, CT prevents soil sealing [32]. It is also well-known that increasing soil carbon sequestration by reducing tillage intensity (frequency and depth) improves soil biodiversity [33]. The presence of a vegetation cover due to CT increases soil biodiversity and can provide a habitat for arthropod predators and parasitoids, promoting biological control of pests and pathogens [16,34,35]. In addition, the build-up of soil organic matter derived from below-ground plant biomass inputs provides food and energy sources for microorganisms, favouring microbial growth and activity. Microorganisms decompose organic matter and increase nutrient availability for crops [33,36]. The presence of plant cover improves soil structure, porosity, aggregate stability, and water infiltration compared to bare soil. Therefore, CT also influences water regulation through the increase in soil water infiltration, which in turn fosters groundwater storage and lessens surface runoff, improving the availability of water for crops [16,37]. Moreover, it has been demonstrated that CT reduces soil erosion by water and wind due to the development of a vegetation cover. Reductions in runoffs of between 30% and 65% and in erosion of between 63 and 80% with decreasing tillage intensity have been observed worldwide [38–40], which ultimately lead to reductions in the nutrient losses resulting from erosion. Additionally, CT has been proven to improve nitrogen availability [35,41]. Other beneficial effects of CT are that the presence of the plant cover enhances soil aggregation, thus improving the protection of SOC against erosion, tillage operations, and abrupt soil temperature and moisture fluctuations [42].

Generally, CT positively impacts crop yields because the enhancement of the organic matter inputs into the system improves water infiltration and storage capacities and the availability of nutrients in soils [43]. In any case, the benefits are sometimes observed a few years after adoption, and the magnitude of the impacts of CT on crop yields depends on pedoclimatic conditions, crop types (arable or woody), and management practices (e.g., rotations, irrigation, and fertilization) [44,45].

In relation to climate change mitigation and adaptation, CT generally reduces GHG emissions compared to conventional (intensive) tillage systems. First, the reduction in the number of passes per year not only mitigates direct $CO_2$ emissions from the machinery but also prevents the peaks of $CO_2$ emissions from soils that typically occur after tillage operations [42]. Second, since CA usually includes improvements in N management, $N_2O$ emissions from soils are reduced or, at least, a decrease in yield-scaled $N_2O$ emissions is achieved [46,47]. Moreover, the presence of plant cover all year round not only protects the soil against erosion while improving its water retention capacity but also increases its buffer capacity against temperature extremes, making soils more resilient to extreme rainfall events, droughts, and warming [33,39,42].

CT has also been proven to have other socio-economic benefits, such as: (i) fuel, fertilizer, and pesticide savings; (ii) reducing erosion and flood risks and associated damage to infrastructure; (iii) sustainable preservation of cultural landscapes; and (iv) maintenance of crop yields, agricultural activity, and long-term employment, contributing to maintaining the local population in rural areas [16,48]. The impact of SOC sequestration goes beyond improving soil properties and synergizes with other biophysical ecosystem services, positively affecting further non-material ecosystem services—or nature's contribution to people—by providing learning opportunities and inspiration, as well as physical and physiological experiences, and supporting identities [49,50].

3.1.4. Possible Drawbacks and Recommendations

Various biophysical, technical, social, economic, cultural, and political barriers can restrict the adoption of CT worldwide. For any given location, the success or failure of CT will depend on one or more of the following factors.

(a)    Biophysical barriers

Local pedoclimatic conditions (soil type—particularly its organic matter content and texture—rainfall amount and distribution, and temperature), together with slope, crop type (arable vs. woody crops, water requirements, growing period, rooting characteristics), and management (rainfed vs. irrigated, conventional vs. organic), determine the viability of field operations, as well as whether crops will be established and their yields. For example, in arid and semiarid regions, the adoption of CT may be hampered because of competition for water and nutrients between the plant cover and the main crop [51,52], this effect being more visible when aridity and temperature increase [43]. On the other hand, in water-logged and heavy-clay soils (e.g., rice fields), reduced tillage is hampered. Moreover, on-site and off-site soil and water contamination problems may arise if pesticides and inorganic fertilizers are applied in high doses [16]. It is also important to note that the positive impacts of reducing tillage operations (i.e., reducing direct emissions from the activities and fostering sinks via SOC sequestration) can be counterbalanced by the increase in soil $N_2O$ emissions in cases where higher doses of inorganic fertilizer are needed, as has been pointed out in a recent meta-analysis [53]. However, the results vary depending on: (1) the duration of the experiment and (2) the management type (e.g., fertilizer type and application rate, use and type of spontaneous and planted cover crops, and crop residue management); therefore, no general conclusions can be drawn. In this regard, CT needs to be accompanied by wise management of nitrogen and weeds.

(b)    Technical barriers

One technical barrier that hinders the wider adoption of NT practices in Europe is the unavailability of proper machinery, such as direct drilling machines or machinery to manage crop residues or cover crops [54]. For instance, the adoption of direct sowing is still a challenge for many crops—in particular, small-seed crops—in silty soils prone to crust and heavy compaction in the topsoil [55]. Due to their weight, direct drilling machines may cause soil compaction, hampering the germination of seeds and the effective establishment of seedlings [56]. Although machines can be adapted to specific soil conditions to reduce soil compaction, the increase in purchase costs makes them unaffordable for farmers. In addition to these technical constraints, one important limitation is the heavy dependence on herbicides and pesticides, which can lead to severe pollution of soil, water, and biodiversity resources. To overcome this problem, the development of cheap alternative methods for weed control is pivotal. In this regard, wise management of ground cover and cover crops (i.e., selection of species and varieties to combat weeds, promotion of mechanical instead of chemical termination, leaving plant residues on the soil surface, combining CT with other practices to control weeds, etc.) is recommended.

(c)    Economic barriers

The absence of financial incentives or subsidies to motivate farmers or compensate them for possible yield losses restricts the adoption of CT practices in many regions. Generally, yields are reduced in the short term, but this trend can be reverted in the long term, especially if CT is adopted in combination with other practices (e.g., addition of organic or green manure [57]). As already mentioned, CT normally encompasses the use of agrochemicals (pesticides and mineral fertilizers), resulting in increased costs that farmers cannot afford [58]. In many cases, investments in adapted machinery are necessary but not affordable by farmers because of limited finance and access to capital for implementation. Uncertainty about the development of policies and market fluctuations, together with internal farm factors (such as farm size, debt, tenure, and family status), are other important barriers to overcome.

(d) Social, cultural, and political barriers

Lack of access to appropriate technologies, practices, and equipment is a major barrier in many countries [48]. Moreover, there are other many factors that hinder the adoption of CT, such as farm size and type, the availability of a power source, family structure and composition, the labour situation, access to cash and credit facilities, peer pressure, the degree of autonomy in choosing and implementing results, and community support [59,60]. The main cultural factors that hamper the adoption of CT are lack of awareness among farmers, lack of innovativeness, lack of motivation, and lack of understanding of the agroecosystem [61,62]. In some regions, CT is in conflict with an important cultural symbol of hard work, as tillage is generally believed to symbolize a hard worker, and with the social recognition that a field properly ploughed is "clean" [56,62].

The objectives and priorities of each government will determine how agriculture is managed at the regional and national levels. Lack of economic incentives and support from governments, including subsidies [60]—and, in particular, the lack of strictness in legislation and standards [63]—are the main reasons why the adoption of CA, despite its well-known agro-environmental benefits in the long term, continues to fail in many countries. In this regard, carbon schemes and other political initiatives are urgently needed (see Section 6).

*3.2. Permanent Plant Cover*

Permanent plant cover refers to those practices involving the growth of a permanent spontaneous or seeded plant cover within the crop system (intercropping systems) or between periods of normal crop production for soil protection and improvement. In the case of spontaneous plant covers, weeds grow in accordance with the pedoclimatic conditions of the area, and species are typically wild species. When the plant cover is seeded by employing what is known as a cover crop, species are selected for which the products can be harvested for food or feed. They may be leguminous (e.g., vetch) so that the cover crop can help to improve the N content, the crops may be used for forage or human consumption (e.g., rye, rapeseed), or mixtures of two or more species may be employed. Spontaneous plant cover can either be removed with a reduced tillage operation so that the plant residues are quickly incorporated into the soil or left on the soil surface; thus, the incorporation of the C and other nutrients will be slower. When seeded cover crops are harvested, their residues are usually left on the soil surface.

3.2.1. Context of Application

Growing a permanent plant cover in intercropping systems is more commonly found with woody crops, since competition for water and nutrients between the woody crop and the plant cover is lower than in the case of arable crops, for which plant covers are usually adopted between normal crop production periods.

Permanent plant covers can be adopted worldwide. However, in rainfed agriculture, they are highly dependent on the precipitation regime. Thus, in arid climates, water availability conditions can place strong limitations on the growth of a permanent plant cover. Regarding the species, a wide variety of wild or seeded species can be grown; therefore, the species composition of the plant cover should be adapted to the specific pedoclimatic conditions and management practices [64]. In the case of seeded cover crops, economic viability plays an important role [65].

3.2.2. Potential for SOC Sequestration

The immediate effect of protecting soil and improving soil conditions is an increase in SOC content. However, the extent of this increase varies with the type of crop, the pedoclimatic conditions, and the specific management practices. Thus, it can range from 0.27 to 1.03 t C ha$^{-1}$ yr$^{-1}$ (Table 3).

**Table 3.** Summary of meta-analyses assessing SOC sequestration rates in different locations and climatic zones. Authors' elaboration published in [66].

| Location | Climate Zone | Additional C Storage Potential (t C ha$^{-1}$ yr$^{-1}$) | Duration (Years) | Cropping System | Reference |
|---|---|---|---|---|---|
| Regional | Warm temperate dry | 0.27 | 10.6 | A + W | [27] |
| Global | Arid, temperate, and tropical | 0.56 | 8.5 | A | [67] |
| Regional | Warm temperate dry | 0.43 | 5.6 | AC + W | [43] |
| Regional | Warm temperate dry | 1.01 | 6.7 | PC + W | [43] |
| Global | Temperate and tropical | 0.32 | 11.9 | A | [68] |
| Regional | Warm temperate dry | 1.03 | 7.7 | W | [1] |

A—arable crops, W—woody crops, AC—annual cover crops, PC—permanent cover crops.

The highest values were achieved for woody crops under warm temperate conditions at around 1.0 t C ha$^{-1}$ yr$^{-1}$, whereas this figure was between 0.3 and 0.6 t C ha$^{-1}$ yr$^{-1}$ for arable crops. This was mainly due to the lower soil disturbance when the plant cover is grown in the inter-row area of woody crops than in the case of arable crops. Average SOC sequestration rates are typically higher in low-duration experiments, when SOC levels are closer to the equilibrium, than in longer ones [69].

### 3.2.3. Co-Benefits

Cover crops and spontaneous plant covers have been reported to not only increase SOC content but also improve other physical (e.g., aggregate stability, water infiltration, and bulk density) [70], chemical (e.g., N, P, and K contents) [71,72], and biological (e.g., microbial diversity, abundance, and activity) properties [73,74], leading to a decrease in the effects of wind and water erosion [70,71] and to higher and more stable yields [75]. SOC increases and improvements in other chemical properties are especially visible with spontaneous plant covers, where the biomass is left on the soil surface or incorporated into the soil with reduced tillage. Cover crops are harvested and eventually used for animal feed or biofuel production [76].

Another benefit of planting cover crops is the weed control resulting from the competition for light, water, and nutrients or the release of allelopathic exudates [75,77]. Cover crops reduce weed density and biomass during the growth of the subsequent cash crop by 10% and 5%, respectively [69]. Weed competition among winter and early-season weeds has been found to have an important role during cover crop growth. On the other hand, growth of spontaneous plant cover in orchards can improve biodiversity and, thus, pollination services and pest control [78].

For cover crops, the diversity of species is also a key driver for the delivery of ecosystem services. However, it becomes a problem when it comes to selecting which species should be planted and how they should be mixed. Indeed, in addition to species diversity, the functional complementary between species is of high importance when mixing plant species. Thus, objective criteria for the selection of species with functional complementary and, thus, the maximization of the delivered ecosystem services have been established [79].

However, it is not only biophysical (provisioning, regulating, and supporting) services that are improved but also cultural and economic ones. The improvement in the soil properties and the competition with weeds in the case of cover crops can lead to reductions in inorganic fertilization and pesticide application, thus leading to lower dependence on external inputs and positive effects on human safety [71]. On the other hand, these better soil conditions might lead to higher and more stable yields [80] because of the increase in the soil resilience. Moreover, in the case of spontaneous plant covers in orchards, improvements to landscape quality (e.g., rural aesthetics) must be considered, as well as other derived socio-economic benefits (e.g., ecotourism and recreational activities) [81].

3.2.4. Possible Drawbacks and Recommendations

The main trade-off from cover crops might be the GHG emissions associated with the decomposition of the organic matter [71], which can be mitigated by using cover crops with low C:N ratios and minimizing the tillage intensity [82,83]. However, these extra emissions might take place only after removing the plant cover. In this context, it has been found that, in olive orchards with spontaneous plant cover and conventional management (weeds controlled with herbicides), the overall amount of $CO_2$ emissions was negative in both treatments (i.e., sinks) but, in the case of the spontaneous plant cover management, the $CO_2$ uptake was double that for the conventional management ($-140$ and $-70$ g C m$^{-2}$ yr$^{-1}$, respectively) [84]. This was mainly due to the increase in the photosynthesis of the plant cover during the growing season, which offset the $CO_2$ emissions after the removal of the plant cover.

In addition to the direct effects from the different management practices on ecosystem services, trade-offs between the different ecosystem services affected by the management practices should be considered [71]. For instance, two positive impacts of cover cropping are the increase in seed production and the increase in faunal activity. However, increases in granivorous faunal activity increase seed predation and, therefore, the subsequent cover crop growth may be negatively affected. These authors suggested applying various tillage activities that could help to control the populations of these insects. Another trade-off is the increase in the soil N content from leguminous cover crops that improve nutrient cycling but, at the same time, may stimulate nematode populations and weed abundance [85,86]. Therefore, determining the right mixture between legume and non-legume species and the right tillage activities could be a way to mitigate these trade-offs between ecosystem services and, at the same time, maintain yields [71]. Cover crop mixtures represent an optimal way to overcome some of these trade-offs [87–89]. Nevertheless, these trade-offs are less common with woody crops under spontaneous plant covers where different and adapted species appear and, therefore, greater self-regulation is achieved.

*3.3. Crop Diversification*

Crop diversification (CD) is a farming system that encourages the cultivation of different plant species in the same field as opposed to monoculture farming [90,91]. There are different options for implementing CD, such as crop rotations (at least two crops in different years), multiple cropping (different crops grown in succession during the same year), and intercropping (crops grown together on the same field). In intercropping, crops can be planted in alternate rows and harvested together (row intercropping) or in wide rows and mechanically harvested separately (strip intercropping), or they can be sown together (mixed intercropping); i.e., with no separation between rows or strips.

In addition to allowing a higher number of crops to be grown and alternated on a field, CD has several objectives [7]:

(a) Covering and protecting the soil from climatic agents in a continuous and effective way;
(b) Maintaining and improving soil structure through the action of the root systems of the plants;
(c) Stimulating biological activity in the soil and eliminating periods with no crop cover;
(d) Limiting environmental risks due to nitrate leaching, erosion and surface runoff, and loss of biodiversity.

3.3.1. Context of Application

CD can potentially be applied worldwide, but barriers to its adoption can emerge from biophysical constraints and cultural and socio-economic factors. In arid and semiarid environments, the climate is warm, and low rainfall limits the cultivation of summer crops if irrigation cannot be supplied; thus, cropping systems are mainly based on winter crops, such as cereals and pulses. Conversely, in cold and wet environments, cropping systems are mainly based on spring–summer crops, since low temperatures, snow accumulation, and the surplus of water during the autumn–winter months can restrict crop growth.

### 3.3.2. Potential for SOC Sequestration

In a global data analysis of 97 paired treatments from long-term experiments (Figure 2), the results indicated that enhancing rotation complexity (i.e., changing from monoculture farming to continuous rotation cropping or from crop–fallow to continuous monoculture or rotation cropping or increasing the number of crops in a rotation system) increased SOC by $0.15 \pm 0.11$ t C ha$^{-1}$ on average [92]. However, changing from continuous corn to corn–soybean rotation did not help sequester C ($-0.19 \pm 0.19$ t C ha$^{-1}$) due to the lower residue return and C input in the rotation compared to the corn monoculture. This result was consistent with findings from the Midwestern USA [93] reporting a SOC loss of 0.15 t C ha$^{-1}$ for corn–soybean rotations with NT and residue incorporation. Not considering corn–soybean rotation, the average SOC sequestration rates were $0.20 \pm 0.12$ t C ha$^{-1}$ and $0.16 \pm 0.14$ t C ha$^{-1}$ under conventional tillage and $0.26 \pm 0.56$ t C ha$^{-1}$ with NT rotations. Rotations with grass, hay, or pasture increased SOC by $0.19 \pm 0.08$ t C ha$^{-1}$ on average. Decreasing the fallow period in wheat experiments (e.g., changing from a wheat–fallow rotation to a wheat–wheat–fallow rotation) and rotating wheat with one or more different crops (e.g., wheat–sunflower or wheat–legume rotations) increased SOC by $0.51 \pm 0.47$ t C ha$^{-1}$ and were more effective than changing from wheat–fallow to continuous wheat farming ($0.06 \pm 0.08$ t C ha$^{-1}$).

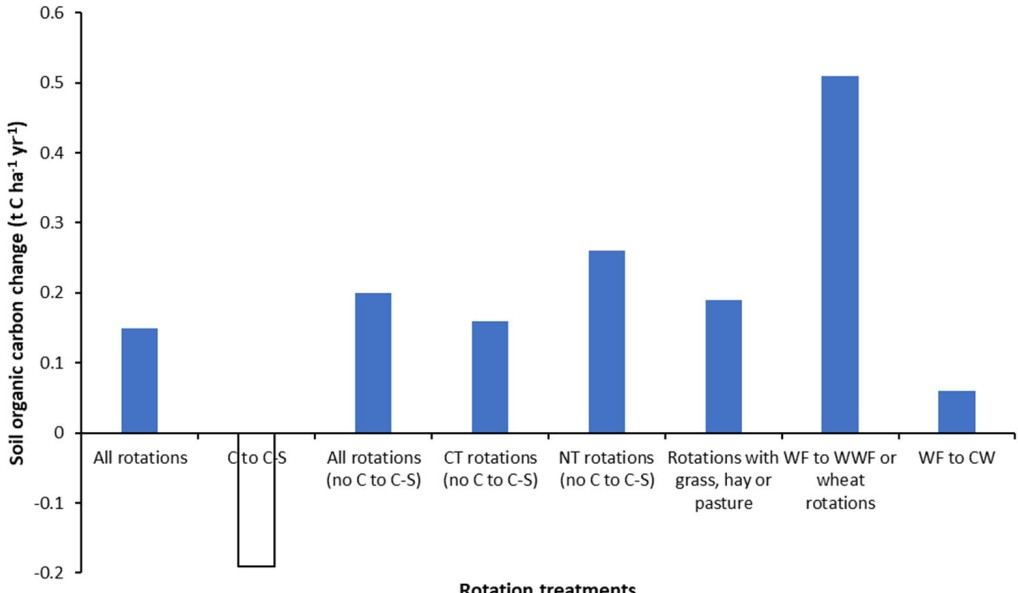

**Figure 2.** Effects of rotation complexity on SOC change. C to C-S—continuous corn to corn–soybean, CT—conventional tillage, NT—no tillage, WF to WWF—wheat–fallow to wheat–wheat–fallow, WF to CW—wheat–fallow to continuous wheat. Authors' elaboration based on [92].

The effects of the number of crops included in a rotation were investigated in a meta-analysis including 122 studies with 454 observations [94]. The results indicated that total soil C (TC) increased by 3.6% on average with the addition of one or more crops in the rotation compared to a monoculture. TC increased by 1.9% with two crops in the rotation, 7.5% with three crops, and 3.7% with four crops. The highest TC responses to rotation were found for soybean (11%), sorghum (7.9%), and wheat (2.9%) monocultures, but rotations did not increase soil C compared to corn monocultures. The introduction of a cover crop in the rotation increased TC by 7.8%, but no significant effect was found in rotations without cover crops. Mean annual temperature and rainfall were correlated positively with rotation effects on TC.

A recent data analysis (304 paired samples) assessed SOC content as affected by CD in different European regions [90]. SOC increased by 18% compared to the control treatment (no rotation/no legumes) when adopting more complex rotations and introducing legume crops. In contrast, SOC decreased in long rotations without legumes (6%) and in short rotations with legumes (3–5%). Furthermore, SOC increases were greater in semiarid climates (11%) compared to humid and sub-humid conditions. The results also indicated greater SOC increases (28%) 2–10 years after adopting CD; in contrast, SOC changes were showed a decreasing trend after 11–20 years (6%) and became definitely negative (−6%) in sites where CD had been adopted for very long time periods (>20 years), showing that a steady-state condition was reached.

Table 4 illustrates the Spearman rank correlation analysis for changes in SOC and several pedoclimatic and predictive variables of CD (e.g., rotation, tillage, fertilization, and residue management). The significant negative coefficient found for the duration of the experiment in years (−0.45) indicates that SOC changes were greater when CD had been established more recently, while significant positive coefficients for SOC changes were found for crop rotations ≥ 3 years (0.61), legumes in the rotations (0.60), conventional tillage (0.22), and the removal of crop residues (0.58). Negative coefficients were also found for no tillage (−0.32), residue incorporation and mulching (−0.31 and −0.33, respectively), mixed fertilization (−0.16), autumn–winter cereals in the Southern Mediterranean region (−0.45), and clay and loam textures types (−0.28 and −0.41, respectively). Positive coefficients were found for semiarid climates (0.16), autumn–winter cereals of the Northern Mediterranean region (0.17), and sandy clay loam soil textures (0.61).

**Table 4.** Spearman rank correlation coefficients (rs) for changes in SOC content and the predictive variables for crop diversification. Authors' data based on [88].

| Variable | Coefficient |
| --- | --- |
| SOC control * | 0.20 |
| Rotation every 3 years * | 0.61 |
| Years * | −0.45 |
| Legumes * | 0.60 |
| Cover crop | −0.04 |
| Conventional tillage * | 0.22 |
| No tillage * | −0.32 |
| Mineral fertilization | 0.08 |
| Mixed fertilization * | −0.16 |
| Organic fertilization | 0.11 |
| Residue incorporated * | −0.31 |
| Residue mulched * | −0.33 |
| Residue removed * | 0.58 |
| Semiarid * | 0.16 |
| Subhumid | −0.14 |
| MedNCerAw * | 0.17 |
| MedSCerAW * | −0.45 |
| BorFodMix | 0.11 |
| Clay * | −0.28 |
| Loam * | −0.41 |
| Sandy clay loam * | 0.61 |

The asterisks (*) indicate both positive and negative correlations with significant coefficients at $p < 0.05$ above rs = |0.15|.

### 3.3.3. Co-Benefits

Adopting CD provides further benefits for soil properties. It can lead to an overall improvement in soil structure resulting from the aggregation of mineral particles and organic materials. Germination and rooting of crops are facilitated by the higher resistance of the soil aggregates to physical stress [95,96], and better soil aggregation also improves

carbon storage due to the physical protection of organic materials [97]. Furthermore, soil crusting and erosion are avoided [98].

CD also improves soil biological properties because different crop species have different C:N ratios (residue qualities), which enhance the activities of different types of soil microorganisms. Furthermore, adopting CD is more effective if coupled with other CA practices (e.g., RT or NT). For example, in the temperate conditions of the northeastern USA [99], adopting NT and cover crops in a crop rotation (maize and perennial grass) system improved active C, respiration, and protein content. Similarly, a rotation system with wheat and forage crops enhanced the microbial biomass carbon under both rainfed and irrigated conditions (by 0.4 and 14.9%, respectively) in comparison to continuous wheat cropping [100]. Furthermore, soil microbial richness and diversity were increased by C (15.11 and 3.36%, respectively) [36].

Crop rotations also spread out the need for labour, reduce equipment costs and peak labour demand, smooth out price fluctuations in markets, and increase local community interaction for labour [101]. However, the possible lack of a market for the alternative crops adopted for CD can represent an economic barrier [102].

Crop yields are generally higher if crops are cultivated after unrelated species, which is known as the break-crop effect. Cultivating a break crop increased wheat yield from 0.5 to 1.2 t ha$^{-1}$, particularly when wheat was cultivated after legumes (e.g., faba beans and chickpeas) [103]. It has also been reported that longer and more complex crop rotations increased yields by 12% compared to monocropping systems, and the increase was lower (5%) for the shortest rotations (2 years) [104].

Nitrogen fertilization is the main agricultural contributor of soil nitrous oxide ($N_2O$) emissions to the atmosphere [105]. Since legumes fix the atmospheric N that is available for plant nutrition, their adoption in CD implies the supply of lower amounts of N fertilizers, thus reducing the $N_2O$ emissions and mitigating their global warming potential. Some examples have been reported from the Northern Great Plains of North America [106], France [107], and Australia [108].

### 3.3.4. Possible Drawbacks and Recommendations

Local pedoclimatic conditions (e.g., rainfall and soil texture) can limit the cultivation of some crops in specific environments, and the market opportunities for each crop included in the diversification scheme must be considered. As already mentioned, cereals (e.g., wheat) have high nitrogen requirements; thus, including $N_2$-fixing legumes in the planned rotations can both increase cereal yields and limit nitrogen losses through $N_2O$ emissions and leaching [109].

Farmers often perceive CD negatively because they fear possible decreases in yields and economic benefits. However, the crops adopted in diversification should be those already grown locally as monocultures, since they have been proven to be suitable for the soil and climatic conditions and provide good yields. Therefore, farmers only need to learn how to use them in rotations, multiple cropping, or intercropping systems. However, not all farmers are skilled in CD. Therefore, providing adequate training to agricultural technical advisors is crucial to successfully disseminate diversified cropping systems among farmers [102,104].

## 4. Processes

### 4.1. The Soil Carbon Balance and Different Processes of SOC Loss in Agroecosystems

Soil carbon storage in agricultural systems is governed by the difference that exists between the carbon inputs from crop biomass (roots plus aboveground crop residues after harvesting and pruning) and any endogenous (e.g., ground covers) and/or exogenous (e.g., manure, compost, sludge, and/or cover crops) organic matter added to the soil, on the one hand, and the carbon outputs, as affected by erosion, leaching, and the decomposition and mineralization of plant material and organic matter at both short- and long-term scales, on the other hand (Figure 3).

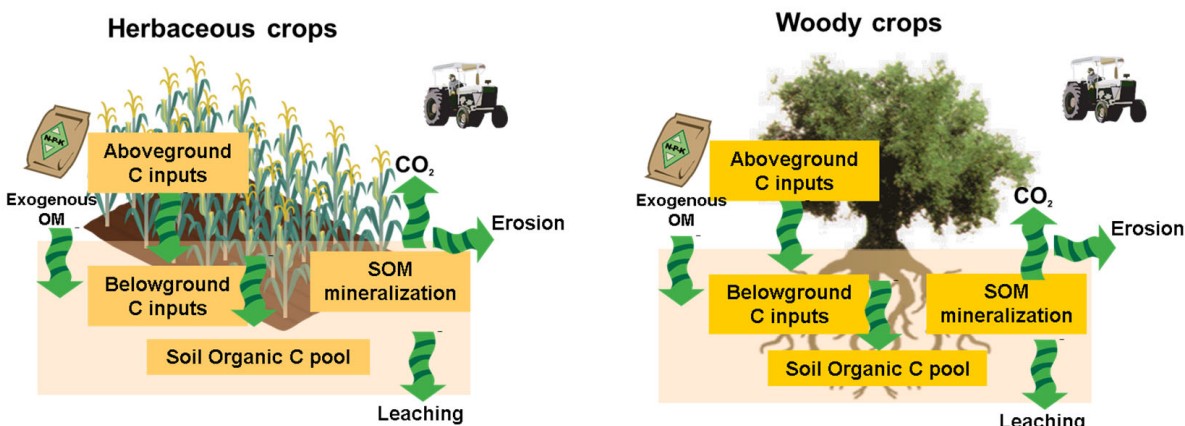

**Figure 3.** The soil carbon balance in herbaceous (**left**) and woody (**right**) cropping systems depends on the difference between carbon inputs and outputs throughout the agroecosystem, as explained in the text. © María Almagro.

To date, most studies focused on the assessment of the effects of CA practices on the soil carbon balance have been based on empirical data derived from field plots and laboratory assays, and scaling up the potential for carbon sequestration from the farm to the global scale still remains a challenge for the scientific community. An excellent indicator of the effectiveness of a certain CA practice is undoubtedly the increase in SOC content, given its well-known agro-environmental benefits and its potential for climate change mitigation [16,110,111]. However, further research and a robust monitoring, verification, and reporting framework are still needed to increase carbon gains and address the limitations of SOC sequestration [112]. In this regard—and given the huge uncertainty associated with SOC estimations at the farm level, particularly in the short term—it is recommended that long-term monitoring programs assess SOC changes a decade after the implementation of the CA practice. However, shorter-term assessments may be needed to guide policy debates and decisions. To address this, estimations of the carbon gains and losses occurring throughout the agroecosystem when a certain management practice is adopted can be used to anticipate decisions concerning agriculture management based on early assessments of SOC net balances. In other words, if the annual amounts of carbon entering the soil due to the addition of organic amendments, the implementation of cover crops, and crop residue retention exceed the carbon losses through erosion, leaching, and decomposition, the SOC balance will be positive and SOC sequestration will be achieved in the short term. However, if the opposite occurs, organic carbon will be lost from the soil system. Notably, each of the described process causing carbon gains or losses at the agroecosystem level contributes differently depending on the specific site (i.e., local climate, soil and crop type, slope, etc.) and management (i.e., rainfed vs. irrigated regime, low- vs. high-input systems) conditions, which, in turn, drive the direction and magnitude of the impact on the net soil carbon balances.

*4.2. Erosion by Water*

Soil water erosion refers to lateral movement of soil downhill caused by significant rainfall events. Soils in natural ecosystems are considered to be under steady-state conditions, as the loss of soil material due to erosion from a given area is approximately balanced by the formation of soil as a result of weathering [113]. However, manmade actions, such as intensive agriculture, deforestation, and soil sealing, have increased soil erosion rates by 10–40 times globally, causing on-site and off-site negative environmental impacts [114]. On-site negative impacts include the loss of carbon and nutrients, such as nitrogen and phosphorus, from the topsoil, reducing soil fertility and crop productivity and causing land degradation and desertification, as recognized by the Soil Thematic Strategy [115] of the European Commission. Off-site effects include sedimentation of reservoirs, eutrophication

of water bodies, and damage to infrastructure [116]. Although normally overlooked, this process is relevant for the soil carbon and nutrient budgets of agroecosystems. Moreover, soil erosion interacts with other relevant processes, such as organic matter decomposition, SOC sequestration, and net primary productivity [117–119].

A global review of empirical data indicated that soil erosion rates from conventionally ploughed agricultural fields (~1 mm yr$^{-1}$) are, on average, one to two orders of magnitude greater than the rates of soil formation, erosion under native vegetation, and long-term geological erosion [113]. These results prove that conventional tillage is unsustainable, particularly in Mediterranean regions where extreme high-rainfall events can cause great soil losses through erosion in a few hours [120]. The same study also indicated that NT systems produce erosion rates much closer to soil formation rates (~3 mm century$^{-1}$, as reported in [121]), highlighting their contributions to soil conservation, mitigation of climate change through the retention of carbon in soils, and sustainable agriculture. However, the success of the various CT practices that can be adopted depends on the local environmental conditions (e.g., soil type, climate, and management practices), as well as the socio-economic, cultural, and political contexts [48,54]. Moreover, the combination of two or more CA practices is generally more effective than the adoption of a single agricultural practice. A comprehensive overview of different soil and water conservation practices in Europe and in the Mediterranean Basin indicated that annual runoff and soil loss rates can be reduced by 20–74% if CA practices are adopted [122]. However, this review also concluded that vegetation management practices (such as the adoption of cover crops and mulching) were the most effective in reducing annual runoff and soil loss rates, followed by mechanical techniques (such as terraces, contour bounds, and geotextiles) and soil management practices (such as NT, RT, contour tillage, and soil amendment), which were the least effective in controlling runoff and erosion. These results highlight the importance of ensuring permanent soil cover in order to reduce soil erosion rates globally. Nevertheless, the more erosion-prone conditions are (i.e., erodible soils, steeper slopes, areas with low-frequency occurrence of high-intensity rainfall events), the more effective these CA practices will be in reducing runoff and soil erosion rates.

*4.3. Decomposition*

Decomposition refers to the physical, biological, and chemical breakdown and leaching of soluble compounds of plant biomass residues (leaves, shoots, and roots) and soil organic matter (SOM), along with the subsequent mineralisation and humification of organic compounds [123]. Decomposition is one of the most important processes in terrestrial ecosystems because it controls SOM formation and the release of organic nutrients and energy for plant growth and soil microorganisms [124,125]. Moreover, it is a major component of carbon and nutrient cycling in ecosystems and a key driver of soil fluxes of carbon dioxide ($CO_2$), methane ($CH_4$), and nitrous oxide ($N_2O$) into the atmosphere. It is estimated that 60 Pg of $CO_2$ is emitted annually by the decomposition of plant litter and SOM [126].

Among the main environmental drivers of plant litter decomposition are temperature, moisture availability, the chemical composition of plant litter, and the soil biotic community structure and activity, which altogether control carbon and nutrient sequestration efficiency in agricultural soils [127]. However, despite the importance of this process, it is still unclear which environmental factors control it and how we can ensure that a significant proportion of the decomposed plant material is returned to the soil instead of released into the atmosphere in the form of $CO_2$ and $N_2O$. This is particularly important in arid and semiarid environments, such as in many Mediterranean regions, where solar ultraviolet (UV) radiation has been identified as a significant driver of plant litter decomposition [128,129]. In this process—known as photodegradation—solar radiation directly breaks down organic matter components, releasing $CO_2$ and other gases and, thus, promoting the direct loss of carbon and nutrients from ecosystems into the atmosphere without incorporation into the SOM pool [128]. Photodegradation is a complex process in which several abiotic (e.g., ambient temperature and moisture, plant residue chemical composition) and biotic (e.g.,

local microbial community response to solar UV radiation) factors interact; as a result, its net effect on plant litter decomposition can be positive, negative, or neutral [130,131]. A recent meta-analysis showed that exposure of plant residues to solar radiation sped up decomposition by 23% [129]. Therefore, photodegradation can negatively impact the SOC content and fertility level in semiarid agricultural soils if crop residues are not wisely managed in these environments. In other words, if photodegradation dominates the decomposition process under certain environmental conditions, facilitating the direct loss of carbon and nutrients from plant residues into the atmosphere, then it may be desirable to incorporate them into the soil through RT (rather than leaving them on the soil surface as mulching) to promote SOC sequestration and fertility.

On the other hand, under arid and semiarid conditions, the decomposition of plant residues mediates soil inorganic carbon (SIC) dynamics and can, therefore, change the net carbon balances of agricultural systems, converting them into sources or sinks depending on their management (rainfed vs. irrigated, the chemical composition of crop residues, and crop residue incorporation into the soil vs. mulching) and local conditions (mainly mean annual precipitation and soil pH). Specifically, the fate of the released $CO_2$ during the decomposition and mineralization of plant residues can lead to formation or dissolution of pedogenic carbonate, leading to its sequestration or to its direct release into the atmosphere, depending on the aridity conditions and soil pH [132].

*4.4. Leaching*

Soil leaching is the downward movement of nutrients (i.e., nitrate, phosphorus, and base cations) and other constituents in the soil profile, such as dissolved organic carbon (DOC) and dissolved inorganic carbon (DIC), following the percolation of rain or irrigation water. This process occurs when the soil pores become filled with water and water moves downward in the soil, hampering the availability of soil nutrients for plants and, therefore, reducing soil fertility and plant yield [133]. Moreover, leaching may cause environmental problems, such as eutrophication, when large amounts of certain nutrients move into ground- and surface water.

Natural ecosystems normally have a point of equilibrium between demand and supply for nutrients, with a closed loop recycling essential nutrients. However, in agricultural cropping systems, the supply of nutrients normally exceeds the demand; therefore, leaching occurs. Global change drivers, such as climate change, land-use change, and agriculture intensification and contamination, affect soil leaching trends.

Leaching of DOC and DIC represents a relatively small but continuous loss of carbon from terrestrial ecosystems. However, only a few studies have estimated carbon losses through leaching in different land-use systems; thus, their contribution to the net ecosystem carbon balance is uncertain [134]. Additionally, climate change may increase the frequency of extreme precipitation events in arid and semiarid regions, leading to increases in SOC losses through both leaching and respiration [135].

For instance, the levels of SOC leached across Europe from forests, grasslands, and croplands have been estimated to be 15.1, 32.4, and 20.5 g C m$^{-2}$ yr$^{-1}$, respectively, which represent 4, 14, and 8% of net ecosystem exchanges, respectively [134]. On the other hand, leaching of biogenic DIC in the same land-use types accounted for lower losses (8.3, 24.1, and 14.6 g m$^{-2}$ yr$^{-1}$ for forests, grasslands, and croplands, respectively) [134].

Additionally, leaching of carbon stored in surface litter and soil layers is considered a main source of DIC and DOC in inland waters [136]. In particular, SIC is more prone to leaching in arid and semiarid regions than SOC via sporadic high precipitation events [135]. This is of great relevance, since SIC stocks and sequestration rates are between two and ten times higher than those for SOC in these areas [137].

## 5. Practices

### 5.1. Conventional Tillage

As mentioned above, SOC sequestration rates vary among studies depending on the local climate conditions, soil characteristics, initial SOC levels, crop type (arable vs. woody cropping systems), previous and current management (rainfed vs. irrigated; low vs. high input systems), and the duration of the experiments. Nevertheless, the increase in SOC in soils is limited in time by the carbon saturation level, and, after a certain point, the rate of accumulation slows down towards a plateau, depending on the soil type, the length of the growing period, and the climatic conditions [26].

It was demonstrated that reducing tillage improved soil aggregation and the protection of organic carbon within the aggregates against erosion or ploughing in two organic rainfed almond orchards under semiarid Mediterranean conditions [42]. The promising results from reducing tillage intensity and frequency were further confirmed by Martínez-Mena et al. [41], who demonstrated that passing from conventional moldboard ploughing at a 40 cm depth (5–7 passes $yr^{-1}$) to minimum tillage at a 20 cm depth (2 passes $yr^{-1}$) in a rainfed cereal field and an organic almond field reduced soil erosion by 65% and 85%, respectively, preventing the carbon losses associated with this process. As a result, SOC stocks at a 30 cm depth increased by 37% and 25%, respectively, in the cereal field and the almond field after six years. On the other hand, however, it was also found that shifting from minimum tillage at a 15 cm depth (twice per year) to NT did not significantly reduce soil $CO_2$ emissions from the soil and negligibly improved SOC stocks (by 1%) after four years in an organic rainfed almond orchard under the same semiarid Mediterranean conditions [42]. Furthermore, crop yields decreased abruptly from the beginning of the cessation of tillage, making this practice unsustainable for local farmers. The failure of NT in this particular case study can be explained by the fact that no fertilization was applied [39], highlighting the importance of adopting NT in combination with other practices, such as addition of organic or green manure, in order to improve N management in semiarid rainfed woody crop systems [138]. Indeed, in an irrigated woody cropping system (i.e., *Citrus limon*) where drip ferti-irrigation was applied together with the addition of pruning residues as mulching, NT was proved to be successful in enhancing SOC stocks, soil aggregation, and OC physicochemical protection at 0–5 cm soil depths after 20 years, thus improving soil structure and halting carbon losses [139]. Nevertheless, given the high spatial variability observed when measuring SOC in agricultural fields, long-term studies are encouraged to assess SOC stock trends over time and thereby estimate average SOC sequestration rates more accurately. For example, SOC was sequestered at a rate of 1 t C $ha^{-1}$ $yr^{-1}$ when shifting from conventional to RT at a 20 cm depth after 10 years in an organic rainfed woody crop system, while a rate of 0.33 t C $ha^{-1}$ $yr^{-1}$ was obtained when shifting from RT to NT at a 15 cm depth under the same conditions [140].

For traditional cereal–fallow rotation and a continuous cropping system with barley under semiarid Mediterranean conditions in northeastern Spain, the results indicated that the adoption of RT (with chisel ploughing at 25–30 cm depths) and NT in formerly conventionally tilled (with mouldboard ploughing at 30–40 cm depths) fields improved soil aggregate formation and stability, as well as the OC content associated with them, after 15 years, particularly in the NT system under continuous cropping [141].

In the north of France, the effects of changing from conventional full-inversion tillage to NT and shallow tillage in combination with different crop management systems (i.e., crop types, residue removal, rotation, and use of catch crops) on SOC stocks were compared after 41 years [142]. The authors demonstrated that tillage and crop residue management had no significant effects on SOC stocks after 41 years at either the formerly ploughed layer (i.e., 0–28 cm) or in whole soil profile (0–58 cm). In the shallow and NT systems, SOC content increased in the surface layer (0–10 cm), reaching a plateau after 24 years, but declined continuously in the subsurface layer (10–28 cm) at rates of 0.42–0.44% $yr^{-1}$. In both the RT and NT systems, SOC sequestration rates increased rapidly during the first four years and then remained more or less constant at average rates of 2.17 and

1.31 t C ha$^{-1}$ yr$^{-1}$, respectively, for the next 24 years, after which they started to decrease. The authors attributed these drops to the water balance in those years, stating that the studied cropping systems sequestered less SOC in wet compared to dry periods, which is the opposite of what occurs under semiarid conditions.

In Lithuania, impacts on SOC sequestration were assessed when shifting from conventional tillage to RT and NT in combination with different fertilization levels in a crop rotation system including winter wheat, spring oilseed rape (*Brassica napus* L.), spring wheat, spring barley (*Hordeum vulgare* L.), and pea (*Pisum sativum* L.) in which crop residue retention was implemented [143]. The SOC sequestration rates were estimated in two long-term (11 years) experiments set up on loam and sandy-loam textured soil. In this study case, NT enhanced SOC sequestration by 5–35% compared to the conventional and RT systems when fertilizer was applied. Specifically, the adoption of NT increased the SOC stocks in the loam soil by 27 and 7% and the SOC stocks in the sandy-loam soil by 29 and 33% compared to the conventional and RT systems, respectively.

The abovementioned contrasting findings highlight the importance of understanding the effects of tillage on SOC sequestration and its interaction with environmental and management factors before drawing conclusions on the potential of CT itself for SOC sequestration.

### 5.2. Cover Cropping

In a meta-analysis of 51 studies and 144 datasets, an average value for the SOC sequestration rate of about 1 t C ha$^{-1}$ yr$^{-1}$ for spontaneous plant covers and cover crops was estimated for Mediterranean woody crops [1]. However, it has been shown that, under Mediterranean climatic conditions, the proportion of non-protected SOC (i.e., available for decomposition and, therefore, not really sequestered SOC) might be between 10% and 50% of the total SOC [144]. Nevertheless, these authors also found that the amount of total SOC in the spontaneous plant cover would be two times higher than that found for conventional management, and statistically significant differences were found for all SOC fractions and the two considered depths (0–5 cm and 5–15 cm), suggesting that the consequences of vegetation cover for SOC extend beyond particulate organic matter and might affect all protected SOC fractions in the first 15 cm.

Similar results were found for grass and legume cover crops in vineyards in Australia [145], where significantly higher concentrations of total, coarse, and fine organic C for the grass–legume mixture and grass-only cover crops were found. However, for the legume-only cover crops, significantly higher values were achieved only for coarse SOC (Table 5). In this study, it was also found that, for mixed cover crops, the total N was generally higher, and extractable N was 75% higher than for the control; furthermore, importantly, plant-available N was 17% greater than with legumes alone. Therefore, a combination of grass and legumes had a positive effect not only on total SOC, including fine particles, but also on the total and plant-available N.

However, even though it is not defined as really sequestered SOM, easily mineralizable organic carbon might play an important role in microbial activity. In an integrated crop–livestock (ICL) system in the USA that included livestock grazing on cover crops and crop residues in agricultural systems, it was found that easily mineralizable SOC and labile C might play important roles in shifting the bacterial community structure and composition in the soil [146]. In particular, these authors found that, in ICL systems, compared to the control, cold-water- and hot-water-soluble carbon levels were increased by 88% and 185%, respectively. These increases in easily mineralizable organic C were associated with significant increases in microbial enzymatic activities (dehydrogenase, fluorescein diacetate, urease, and β-glucosidase activities).

**Table 5.** Means ($\pm$ standard errors) for dependent variables, bulk soil, and coarse and fine SOC fractions by treatment obtained from linear mixed effects models examining the effects of cover crop type on dependent variables at four sites (n = 16). Different lowercase letters represent significant differences between treatment groups ($\alpha$ = 0.05). Authors' elaboration based on [145].

| Cover Crop | SOC Bulk Soil (mg g$^{-1}$) | SOC Coarse Fraction (mg g$^{-1}$) | SOC Fine Fraction (mg g$^{-1}$) |
| --- | --- | --- | --- |
| Grass only | 14.22 $\pm$ 1.22 b | 7.42 $\pm$ 1.43 b | 35.50 $\pm$ 4.74 b |
| Legume only | 13.62 $\pm$ 1.10 a | 6.96 $\pm$ 1.15 b | 32.77 $\pm$ 4.23 a |
| Mixture | 14.64 $\pm$ 1.29 b | 9.36 $\pm$ 1.86 b | 34.56 $\pm$ 4.55 b |
| Control | 11.41 $\pm$ 1.02 a | 5.36 $\pm$ 1.01 a | 30.57 $\pm$ 3.91 a |
| *p*-value | <0.01 | <0.01 | 0.02 |

A clear relationship between cover crops, SOC, and soil biological parameters was also found for Andisols in Japan with arable crops [147]. Results showed that a combination of NT and the use of rye as a cover crop could enhance SOC and soil health parameters (total N, available P, exchangeable K-Mg, CEC, bulk density, soil penetration resistance, and substrate-induced respiration) in soybean crops After a Z-score assessment, these authors found a positive effect from the use of rye as a cover crop, especially for soil biological and chemical features, and it significantly increased the cover crop biomass input

Finally, regarding the SOC dynamics over time and long-term SOC sequestration, a simulation study of SOC dynamics was performed for NT with cover crops (winter cereal) and conventional tillage in a continuous maize system in the USA for the period 1970–2099. The results showed that, in 1970–2018, the SOC gains were 0.22 t C ha$^{-1}$ yr$^{-1}$. However, sequestration rates under climate change were much lower, with gains equal to 0.031 t C ha$^{-1}$ yr$^{-1}$ with NT compared to conventional tillage in the IPCC RCP 8.5 scenarios and lower SOC losses in the case of the RCP 2.6 scenarios of $-0.002$ vs. $-0.017$ t C ha$^{-1}$ yr$^{-1}$ for NT with cover crops and conventional tillage, respectively [148].

*5.3. Crop Diversification*

A recent meta-analysis [149] demonstrated that CD generally improves pollination and pest control, water regulation, carbon sequestration, nutrient cycling, and crop yields, although exceptions to this general trend were also observed. In this context, long-term field experiments (LTEs) can be used as robust research instruments for the study of ecosystem productivity and sustainability because they capture the changes in and relationship between cropping systems, agricultural management, and the fluctuating environment at different time points over long periods. LTEs also make it possible to quantify the effects of CD through crop rotations on SOC storage.

Two monocultures (continuous corn and continuous soybean) and three rotations (soybean–corn, soybean–winter wheat, and soybean–winter wheat–corn) were evaluated in an LTE using conventional tillage (CT) and NT in Canada (Ontario) [150]. After 11 years (Figure 4), for the soybean–corn rotation compared to continuous corn farming, SOC was higher by 9.6 t C ha$^{-1}$ (18.8%) using CT, lower by 18.4 t C ha$^{-1}$ (26%) using NT, and lower by 3.8 t C ha$^{-1}$ (6.3%) on average. For the soybean–winter wheat–corn rotation compared to continuous corn farming, SOC was slightly higher in using (0.4 t C ha$^{-1}$, 0.8%) and lower using NT (1.3 t C ha$^{-1}$, 1.8%) and on average (0.3 t C ha$^{-1}$, 0.5%). For the soybean–winter wheat rotation compared to continuous soybean farming, SOC was higher by 33.8 t C ha$^{-1}$ (74.8%) using CT, 17.5 t C ha$^{-1}$ (28.1%) using NT, and 26.3 t C ha$^{-1}$ (49.5%) on average. For the soybean–winter wheat–corn rotation compared to continuous soybean farming, SOC was higher by 6.3 t C ha$^{-1}$ (13.9%) using CT, 7.3 t C ha$^{-1}$ (11.7%) using NT, and 6.8 t C ha$^{-1}$ (12.8%) on average. The overall results suggest the efficacy of the incorporation of winter wheat in the rotations, adopting soybean–wheat and soybean–winter wheat–corn rotations rather than monocultures based on corn and soybean or soybean–corn rotations.

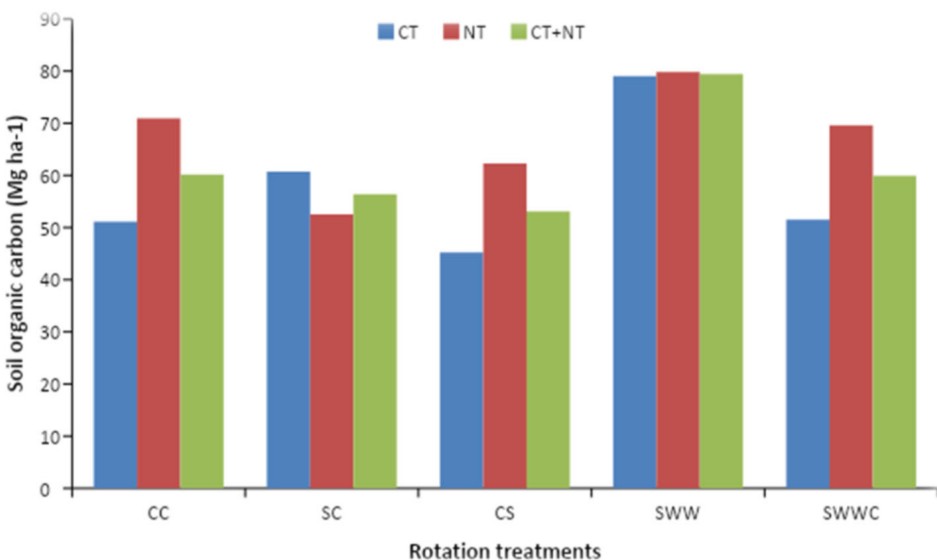

**Figure 4.** Effects of rotation treatments on SOC storage. CC—continuous corn, SC—soybean–corn, CS—continuous soybean, SWW—soybean–winter wheat, SWWC—soybean–winter wheat–corn, CT—conventional tillage, NT—no tillage. Authors' elaboration based on [150].

A study performed in the semiarid Pampean region of Argentina (Buenos Aires Province) over 15 years examined three treatments with and without fertilizer inputs: continuous wheat (WW), 1 year of wheat followed by 1 year of grazing of natural grasses (WG), and 2 years of wheat followed by 2 years of legume (clover, vetch) and grass (barley, oat, triticale) mixtures (WL) [151]. The results demonstrated the positive influence of the inclusion of legumes (WL) on SOC, as well as that of alternate cattle grazing (WG), while continuous wheat showed the lowest SOC storage. Compared to continuous wheat with no fertilization, SOC increased by 3.1 t C ha$^{-1}$ (7.7%) and 3.8 t C ha$^{-1}$ (9.5%) in the WG and WL treatments, respectively. With fertilization (64 kg N ha$^{-1}$ and 16 kg P ha$^{-1}$), SOC increased by 1.8 t C ha$^{-1}$ (4.1%) and 7.6 t C ha$^{-1}$ (17.4%) in the WG and WL treatments, respectively (Figure 5).

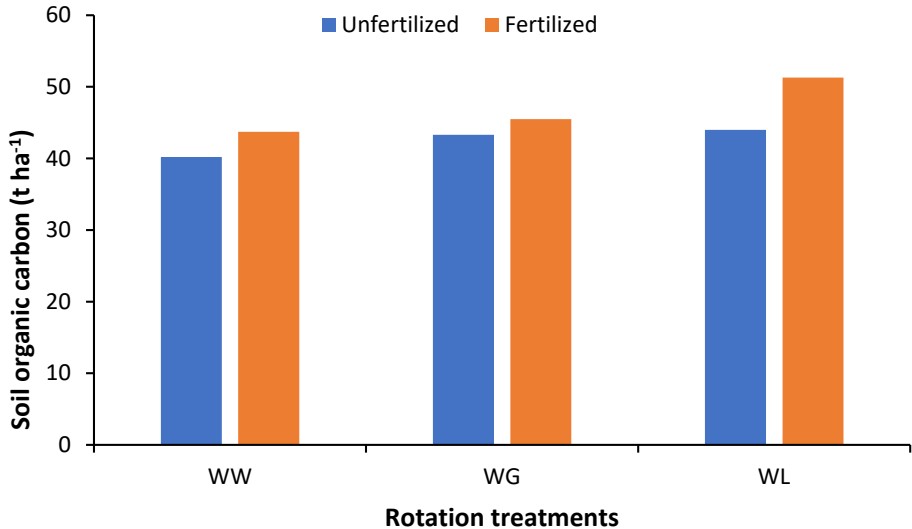

**Figure 5.** Effects of rotation treatments on SOC storage. WW—continuous wheat, WG—1 year of wheat followed by 1 year of grazing of natural grasses, WL—2 years of wheat followed by 2 years of legume and grass mixtures. Authors' elaboration based on [151].

An LTE (18 years) performed in the Western Corn Belt (NE, USA) evaluated three monocultures (continuous corn, continuous soybean, and continuous grain sorghum), two 2 year rotations (corn–soybean and grain sorghum–soybean), and two 4 year rotations (oat+clover–grain sorghum–soybean–corn and soybean–grain sorghum–oat+clover–corn) using three nitrogen fertilization levels (0, low, and high—i.e., 0–90–180 kg N ha$^{-1}$—for corn and sorghum and 0–34–68 kg N ha$^{-1}$ for soybean and oat+clover) [152]. Compared to the corn monoculture (Table 6), SOC increased in the no-fertilizer treatment by 5.6 t C ha$^{-1}$ (1.8%), 9.9 t C ha$^{-1}$ (21.0%), and 7.8 t C ha$^{-1}$ (16.5%) with the corn–soybean, oat+clover–grain sorghum–soybean–corn, and soybean–grain sorghum–oat+clover–corn rotations, respectively. In relation to soybean monoculture, SOC increased in the oat+clover–grain sorghum–soybean–corn rotation by 3.7 t C ha$^{-1}$ (7.0%) and 4.0 t C ha$^{-1}$ (7.5%) with low and high fertilization rates, respectively. Compared to sorghum monoculture, SOC increased in the oat+clover–grain sorghum–soybean–corn rotation by 3.7 t C ha$^{-1}$ (7.0%) and 4.0 t C ha$^{-1}$ (7.5%) with low and high fertilization rates, respectively. SOC increased in the oat+clover–grain sorghum–soybean–corn rotation by 3.3 t C ha$^{-1}$ (6.2%) and 3.0 t C ha$^{-1}$ (5.5%) with low and high fertilization rates, respectively. The overall results indicated that the 4 year rotations with oat+clover crops represented the best option compared to the corn, soybean, and grain sorghum monocultures or 2 year rotations.

**Table 6.** SOC comparisons from 2002 for each rotation and N level.

| N Fertilization | 0 N | | Low N | | High N | |
|---|---|---|---|---|---|---|
| **Rotation** | **Delta SOC** | **%** | **Delta SOC** | **%** | **Delta SOC** | **%** |
| Corn | | | | | | |
| C-SB vs. CC | 5.6 | 11.8 | 3.4 | 6.9 | 1.5 | 2.9 |
| OCL-SG-SB-C vs. CC | 9.9 | 21.0 | 7.0 | 14.2 | 6.2 | 12.1 |
| SB-SG-OCL-C vs. CC | 7.8 | 16.5 | 4.2 | 8.6 | 3.9 | 7.6 |
| Soybean | | | | | | |
| C-SB vs. CSB | −1.4 | −2.5 | 0.1 | 0.1 | −0.7 | −1.3 |
| SG-SB vs. CSB | −5.3 | −9.8 | −3.1 | −6.0 | −4.3 | −8.0 |
| OCL-SG-SB-C vs. CSB | 3.0 | 5.5 | 3.7 | 7.0 | 4.0 | 7.5 |
| SB-SG-OCL-C vs. CSB | 0.9 | 1.6 | 0.9 | 1.7 | 1.7 | 3.2 |
| Sorghum | | | | | | |
| SG-SB vs. CSG | −6.1 | −11.1 | −3.5 | −6.6 | −5.3 | −9.7 |
| OCL-SG-SB-C vs. CSG | 2.2 | 4.0 | 3.3 | 6.2 | 3.0 | 5.5 |
| SB-SG-OCL-C vs. CSG | 0.1 | 0.1 | 0.5 | 0.9 | 0.7 | 1.3 |

CC—continuous corn, CSB—continuous soybean, CSG—continuous grain sorghum, C-SB—corn–soybean, SG-SB—grain sorghum–soybean, OCL-SG-SB-C—oat+clover–grain sorghum–soybean–corn, SB-SG-OCL-C—soybean–grain sorgum–oat+clover–corn. Authors' elaboration based on [152].

## 6. Policy Options

### 6.1. European Union Policy Options

6.1.1. The Soil Thematic Strategy

The Thematic Strategy for Soil Protection is a Communication from the European Commission to the other European institutions [115] involving a 10 year work program for the European Commission. The strategy aims at protecting soil and preserving its capacity to perform its functions in environmental, economic, social, and cultural terms. The strategy includes a legislative framework with four goals: (1) protecting and sustainably using soil, (2) integrating soil protection into national and EU policies, (3) improving knowledge in this area, and (4) increasing public awareness. The proposal for a Directive represents a key component of the strategy, enabling Member States to adopt context-specific measures (e.g., identification of areas at risk of erosion, organic matter depletion, soil compaction, or salinisation) as part of the obligation to adopt programmes of measures addressing causes and impacts.

The European Environment Agency [153] indicated that the lack of a comprehensive and coherent policy framework to protect land and soil is a key gap that may limit the EU's ability to meet future goals. A new policy framework is, therefore, needed, as the 2006 EU Soil Thematic Strategy [115] is no longer adapted to the current policy context and the scientific evidence. This impasse seems close to an end, since the EU Biodiversity Strategy for 2030 [154] provides an update to the 2006 EU Soil Thematic Strategy, aiming to achieve land degradation neutrality by 2030. It highlights the importance of increasing efforts to protect soil fertility, reduce erosion, and increase soil organic matter. Thus, the EU has put soil and land at the core of most of the Sustainable Development Goals (SDGs) of the UN Agenda 2030, particularly SDG 15.3: "combat desertification, restore degraded land and soil, including land affected by desertification, drought and floods, and strive to achieve a land degradation-neutral world by 2030".

6.1.2. The Common Agricultural Policy (CAP) (2023–2027)

In this review, it was shown that increasing SOC with different sustainable management practices results in potential synergies with other ecosystem services. The main European Union instrument used to address sustainability issues in agriculture is the Common Agricultural Policy (CAP). The so-called "Green Architecture" of the new CAP has three specific objectives relating to environmental and climate issues:

1.  Contribute to climate change mitigation and adaptation;
2.  Foster sustainable development and efficient management of natural resources;
3.  Contribute to the protection of biodiversity, thus enhancing ecosystem services and preserving habitats and landscapes.

The new architecture is based on three components, retaining two pillars of the previous architecture (Pillar 1 relating to direct payments) [155]:

(1)  Eco-schemes (voluntary, Pillar 1): direct payments to farmers for the implementation of sustainable management. This is a novel feature of the new Green Architecture, and such schemes can be adapted to the specific needs of the different Member States at the national and/or regional levels. Eco-schemes are intended to play an important role in the new CAP, since 100% of the funding comes directly from the EU and, therefore, no extra funding from Member States is needed;

(2)  Agri-environment–climate measures (AECM) (voluntary, Pillar 2): these measures aim to address environmental and climate challenges using Rural Development Programmes;

(3)  Enhanced conditionality (mandatory, Pillar 1): this component sets out the basic and mandatory requirements that farmers and managers must fulfil in order to receive payments. The requirements refer to the implementation of good agricultural and environmental conditions (GAECs); e.g., maintenance of permanent grasslands, banning of burning arable stubble, implementation of buffer strips in water courses, use of tools for nutrient management, adoption of reduced tillage, avoidance of bare soils in sensitive periods, crop rotation, preservation of a share of the total agricultural area for landscape measures, and banning of the conversion of permanent grasslands in Natura 2000 sites.

The different sustainable management practices addressed in this review relate to the three different CAP components. Importantly, the avoidance of bare soils in most sensitive periods, the use of crop rotations, and the maintenance of a certain ratio of permanent grassland to agricultural areas are practices included in the conditionality, and they involve some of the management techniques previously assessed (e.g., reduced tillage, crop diversification, and cover crops). Fulfilling these requirements would enable farmers to receive the area- and animal-based payments under both Pillars 1 and 2.

6.1.3. The European Green Deal

The European Green Deal [156] was set out by the European Commission in December 2019 with two overall objectives:

- "Transform the EU into a fair and prosperous society, with a modern, resource-efficient and competitive economy where there are no net emissions of greenhouse gases in 2050 and where economic growth is decoupled from resource use";
- "Protect, conserve and enhance the EU's natural capital, and protect the health and well-being of citizens from environment-related risks and impacts".

In order to achieve these goals, the European Commission has established a set of transformative policies addressing different environmental and socio-economic challenges. Most of them are directly or indirectly related to SOC sequestration and preserving or improving soil-supporting functions. They are briefly described below.

Climate Initiatives

The aim of the climate initiatives is to achieve climate neutrality by 2050 [157], enshrining this objective in legislation. To do so, the first European "Climate Law" will be launched, and the targets of reducing GHG emissions by at least 50% compared to 1990 levels by 2030 and towards 55% have been proposed. More specifically, and beyond the creation of a trading system in industry, the aim is to include GHG emissions and removals from land use, land use changes, and forestry [158]. Finally, and in addition to the future efforts in mitigation, the Commission will adopt a new and more ambitious [158] EU strategy on adaptation to climate change, including nature-based solutions [159], where SOC sequestration will play a central role.

From "Farm to Fork": Designing a Fair, Healthy, and Environmentally Friendly Food System

Within the frame of the Green Deal, the EU has developed the "from farm to fork" concept [160] and adapted it to EU biophysical and socio-economic conditions. Thus, the goals of this initiative are "to reduce the environmental and climate footprint of the EU food system and strengthen its resilience, ensure food security in the face of climate change and biodiversity loss and lead a global transition towards competitive sustainability from farm to fork and tapping into new opportunities".

The aim is, therefore, threefold. First, this initiative aims to ensure that the food chain has a neutral or positive environmental impact (preserving and restoring the land-, freshwater-, and sea-based resources on which the food system depends), helping to mitigate climate change and facilitate adaption to its impacts, protect resources (land, soil, water, air) and animal health and welfare, and reverse the loss of biodiversity. Second, the aim is to ensure food security, nutrition, and public health. Third, the initiative aims to preserve the affordability of food while generating fairer economic returns in the supply chain.

Again, SOC sequestration and sustainable management practices in agriculture will play important roles in achieving a more sustainable and fairer European food system. For instance, organic farming is supposed to be promoted through the implementation of the Action Plan on Organic Farming [161], which aims to achieve organic farming on 25% of the total agricultural land in the EU by 2030.

Preserving and Restoring Ecosystems and Biodiversity

The EU has launched the Biodiversity Strategy for 2030 [154] in order to address the biodiversity loss that is threatening food systems and the implementation of healthy and nutritious diets while preserving rural livelihoods and agricultural production in the face of reductions in pollination. The strategy is framed as part of the ambition to "ensure that by 2050 all of the world's ecosystems are restored, resilient, and adequately protected". The strategy links agricultural land management and biodiversity preservation by:

- "Bringing nature back to agricultural land" through the promotion of eco-schemes and results-based payment schemes and by ensuring that the CAP strategic plans

include realistic and robust climate and environmental criteria and targets. These plans should include practices such as organic farming, agro-ecology, and agro-forestry. Furthermore, as also suggested by the EU Pollinators Initiative [162], the overall use of chemical pesticides should be reduced by 50% by 2030. The strategy also aims to restore at least 10% of agricultural areas occupied by high-diversity landscape features (inter alia, buffer strips, rotational or non-rotational fallow land, hedges, non-productive trees, terrace walls, and ponds) in order to enhance SOC sequestration and prevent soil erosion and depletion. Finally, the decline in genetic diversity will be addressed by modifying the marketing rules for traditional crop varieties in order to promote their conservation and sustainable use;

- "Addressing land take and restoring ecosystems" in order to protect soil fertility, reduce soil erosion, and increase SOC through the adoption of sustainable management practices. To promote these practices, the Commission updated the EU Soil Thematic Strategy in 2021. Soil sealing and rehabilitation of contaminated brownfields will be part of the Strategy for a Sustainable Built Environment;
- "Bringing nature back to cities" by calling on European cities of at least 20,000 inhabitants to develop ambitious Urban Greening Plans by the end of 2021 incorporating nature-based solutions;
- "Reducing pollution" through the implementation of the EU Chemicals Strategy for Sustainability [163], the Zero Pollution Action Plan for Air, Water and Soil [164], and the Nutrient Management Action Plan in 2022 [165], which aim to reduce the use of fertilizers by at least 20% and the risks related to and use of pesticides.

The Need for the Integration of CAP Reform and the Green Deal

At this point, it should be clear that there are many links between the new CAP and the Green Deal. However, the new CAP does not directly consider these links, but they are implicitly included in the conditionality and its indicators. However, in the case of eco-schemes, Member States will be in charge of expanding the sustainability of the agricultural sector beyond the requirements of the conditionality throughout the development of CAP Strategic Plans.

The working paper of the European Commission concludes that the CAP reform proposal is compatible with the Green Deal and the associated strategies and initiatives. Nevertheless, to consider these linkages, among others, realistically, it proposes:

- an adequate "no backsliding" principle obliging Member States to be more ambitious in their CAP Strategic Plans than at present regarding environmental and climate-related goals;
- an ambitious system of conditionality to maintain key standards (in particular, for crop rotation, soil cover, and maintenance of permanent grassland and agricultural land devoted to non-productive areas or features);
- mandatory eco-schemes.

The eco-schemes are of critical importance in implementing the climate, air, water, soil, and biodiversity EU goals with regard to country-based and regional specificities. Thus, they should cover those management practices not included in the conditionality. Although some attempts to link soil management with different EU policies have been developed (e.g., Healthy Soils, the EU Soil Observatory, the European Soil Data Centre) [166,167], the reality is that the eco-schemes proposed by many EU countries—and, especially, those related to conservation agriculture—are not ambitious enough, being at best reformative and addressing some specific issues.

Therefore, the challenges of the new CAP for the future are: (1) to integrate the CAP with the Green Deal and its policies and instruments; (2) to increase the ambition of the CAP, particularly the eco-schemes and the enhanced conditionality; and (3) to adopt a systemic view so that the new architecture of the CAP can contribute to the systemic transformation of the agri-food system (Figure 6).

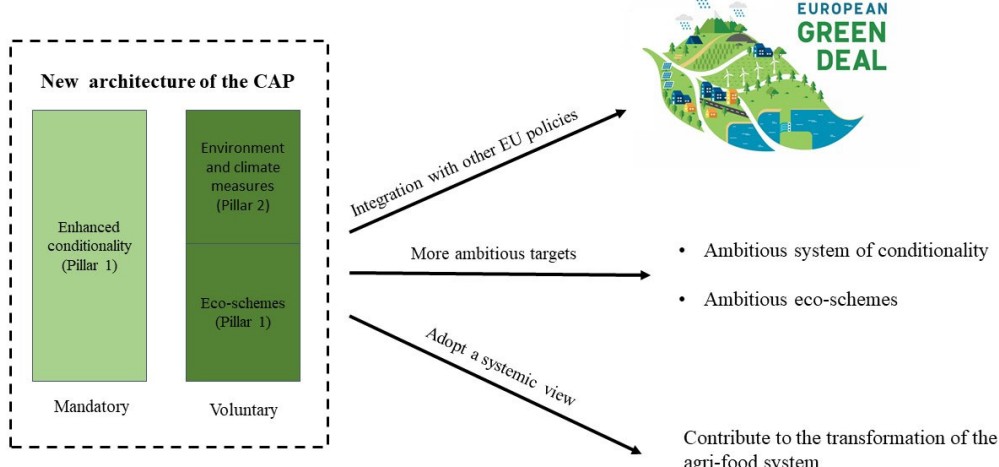

**Figure 6.** The three challenges related to the new architecture of the European Common Agricultural Policy (CAP). Authors' elaboration based on the cited literature.

## 6.2. Other International Relevant Policies

### 6.2.1. The 4 per 1000 Initiative

Climate change is expected to have relevant impacts on SOC dynamics, since the rising atmospheric $CO_2$ concentration could increase biomass production and the crop residues returned to soils. However, increasing temperatures could reduce SOC by accelerating microbial decomposition. The 4 per 1000 initiative Soils for Food Security and Climate, launched by the French Government in 2015 during the 21st Session of the Conference of the Parties of the United Nations Framework Convention on Climate Change in Paris (http://4p1000.org/, accessed on 14 February 2023), is a voluntary action plan aiming at better management of SOC in agricultural soils. The objective is to achieve a 4‰ annual growth rate for SOC stocks in the top 40 cm of soils (i.e., 0.4 per cent per year) as a compensation for the global emissions of greenhouse gases (GHGs) from anthropogenic sources, thus limiting global warming to 2 °C.

Sequestration rates differ between countries and climatic conditions, but a general trend for the relationships between different management practices and SOC accumulation rates has been observed [168]: afforestation—~0.6 t C ha$^{-1}$ yr$^{-1}$, conversion to pasture—~0.5 t C ha$^{-1}$ yr$^{-1}$, organic amendments—~0.5 t C ha$^{-1}$ yr$^{-1}$, residue incorporation—~0.35 t C ha$^{-1}$ yr$^{-1}$, no or reduced tillage—~0.3 t C ha$^{-1}$ yr$^{-1}$, and crop rotation—~0.2 t C ha$^{-1}$ yr$^{-1}$. However, there is a tendency to find higher C sequestration potential (10–30 per 1000) in croplands with low initial SOC stock ($\leq$30 t C ha$^{-1}$) (i.e., high C saturation deficit). In addition, sequestration rates can reach up to 20 per 1000 within the first 5 years after the adoption of sustainable management practices and up to 10 per 1000 after 20 years, then becoming limited to 4 per 1000 after 40 years. However, despite these data, there are still some scientific and policy challenges for the implementation of this initiative [169]:

- The scarcity of scientific data. Research data on rates of SOC sequestration resulting from the implementation of recommended management practices (RMPs) for land use and agricultural management combinations are not widely available;
- The finite capacity of soil carbon sinks. The potential for SOC sequestration in global croplands is finite (0.4 to 1.2 Gt). Thus, SOC sequestration by itself cannot offset all emissions but must be part of a wider set of actions, including the adoption of RMPs that reduce C emissions and enhance C sinks;
- Resource-poor farmers and small landholders who are unable to adopt RMPs because of weak institutional support and poor access to essential inputs. These farmers' degraded and depleted soils need urgent restoration through SOC sequestration and the adoption of RMPs;

- Financial commitments. The adoption of RMPs would require economic resources;
- Permanence. Incentivising the continuous use of RMPs and restorative land uses, as has been undertaken by some successful programs in the EU and USA, is of crucial importance and must be addressed;
- Implementation of the Paris Agreement's 4 per 1000 program. Even though the limitation of global warming to 1.5 °C is required, the word "soil" is never mentioned. Therefore, this is a new challenge for soil scientists and agronomists.

### 6.2.2. Sustainable Development Goals (SDGs)

During the United Nations General Assembly in 2015 [170], the 2030 Agenda for Sustainable Development was adopted. The 2030 Agenda indicates a set of 17 Sustainable Development Goals (SDGs) aiming to "end hunger and poverty, to protect the planet, and to ensure peace and prosperity for all". Each SDG includes specific Targets to be achieved between 2015 and 2030 and to be implemented at the national scale.

Soil sustainable management is directly related to half of the SDGs and indirectly relevant for the other SDGs [171]. The 2030 Agenda adopted specific Targets aiming to restore degraded soils, achieve land degradation neutrality worldwide, and implement agricultural practices to improve soil quality and reduce soil contamination [170].

The stock of SOC has strong interactions with all environmental compartments (e.g., water and air) and supports many soil-derived ecosystem services [172]. Thus, increasing SOC stocks is related to many SDGs and Targets, such as Target 2.4 (improving land and soil quality), Target 15.3 (achieving a land degradation-neutral world), and Goal 13 on climate action, which evaluates climate change and its impacts, aiming to regulate C storage and GHGs [173] and use soil as a C pool [115].

## 7. Concluding Remarks, Future Research Needs, and Policy Recommendations

The widespread adoption of conservation agriculture (CA) principles (i.e., conservation tillage (CT), permanent plant cover, and crop diversification (CD)) could contribute to the mitigation of climate change without compromising food security from local to global scales. However, there are still many scientific knowledge gaps to be filled, as well as biophysical, technical, socio-economic, cultural, and political barriers to overcome, before its adoption can be enabled among farmers worldwide. In this regard, the success or failure of the adoption of any CA practice will depend on the environmental–socio-economic context; therefore, institutional guidance should be planned and created from local to regional scales. Likewise, providing adequate training for farmers to help them implement CA—particularly the adoption of cover crops and CD—and opening up market opportunities for new products are necessary steps in the transition to more sustainable and diversified cropping systems.

CA improves the physical, chemical, and biological properties of soil that are crucial for maintaining soil condition and health, and adopting any of the three principles of CA has beneficial effects on soil organic carbon (SOC), as has been demonstrated worldwide. However, SOC sequestration rates and their co-benefits vary among studies depending on the local pedoclimatic and management conditions, and further research is needed to determine the optimal agricultural management practices within each environmental, socio-economic, and legal context. Attention must also be paid to trade-offs. Furthermore, each of the three CA principles needs to be accompanied by wise and integrated nitrogen and weed control management to ensure sustainable crop yields.

The challenge will be monitoring and verifying that the different sustainable management practices are being applied correctly and assessing how they impact the different ecosystem services. For these purposes, suitable and feasible indicators must be clearly defined. In this regard, the increase in SOC content is an excellent indicator of the effectiveness of a certain CA practice, given its well-known agro-environmental benefits and its potential for climate change mitigation. However, further research and a robust monitoring, verification, and reporting framework are still needed to accurately assess

SOC gains and address the limitations of SOC sequestration. Given the huge uncertainty associated with SOC estimations at the farm level, particularly in the short term, long-term monitoring programs are recommended to accurately assess the SOC gains associated with CA practices. However, short-term monitoring is also needed to guide policy decisions on agriculture management using early assessments of net SOC balances. Likewise, a better understanding of the major processes involved in SOC losses—i.e., erosion, abiotic decomposition, and leaching—and how to curb them is necessary to guarantee the success of CA practices.

Many different strategies, initiatives, and regulations relating to soil ecosystem services have been developed at the regional, national, and international levels, and more will arise in the upcoming years. However, since soil ecosystem services are closely interlinked with other biophysical and socio-economic services, strong coherence between the different initiatives (i.e., the Common Agricultural Policy (CAP), the Farm to Fork Strategy, the Biodiversity Strategy for 2030, the 4 per 1000 initiative, and the Climate Law) is highly recommended. The Sustainable Development Goals (SDGs) could be a suitable framework to achieve this coherence. In addition, considering the new CAP and the recommendations already made by the staff of the European Commission, we encourage Member States to propose ambitious and mandatory eco-schemes in their CAP Strategic Plans involving management practices that aim to increase SOC content and improve and protect soil conditions (e.g., CT, cover crops, diversify cropping systems, etc.) by setting up specific indicators and targets for SOC accumulation in the upcoming years. Nevertheless, trade-offs between increasing SOC storage and GHG emissions should be included in the assessments.

**Author Contributions:** All authors contributed equally in the different phases of the development of the manuscript: Conceptualization, investigation, literature review, R.F., M.A., J.L.V.-V.; Writing—original draft preparation, R.F., M.A., J.L.V.-V.; Writing—review and editing, and visualization, R.F., M.A., J.L.V.-V. All authors have read and agreed to the published version of the manuscript.

**Funding:** This research received no external funding.

**Data Availability Statement:** Not applicable.

**Acknowledgments:** María Almagro acknowledges the financial support from the Spanish Ministry of Science, Innovation and Universities through the "Ramón y Cajal" Program (RYC2020-029181-I).

**Conflicts of Interest:** The authors declare no conflict of interest.

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
