# Peer review of "Conservation Agriculture and Soil Organic Carbon: Principles, Processes, Practices and Policy Options"

_soilsystems, doi:10.3390/soilsystems7010017_

Round 1

Reviewer 1 Report

It is an interesting reviewing work. The authors have presented many valuable things in this manuscript. However, they did not arrange the contents in good manner. Section 3 and 4 should be integrated in one section properly.

 Section 3.1.1: Weed control is generally managed by herbicides or in some cases by crop rotation. Herbicides cause great damage to the soil system. I hope the author can consider carefully and add some biological control or mechanical control.

 Section 3.3: The conversion from conventional tillage to CT increases SOC stocks worldwide. Add references please!

 Sections 3.8 and 3.9 should be integrated with sections 3.3 and 3.4.

 Generally: The structure of section 3 and 4 should be rephrased.

Academic Editor Notes

Dear authors,

Thank you for submitting your manuscript to Soil Systems.

Based on the reviewers’ comments, my decision is accept the manuscript after major revision.

Four reviewers revised your manuscript, and I recommend consider their comments and suggestions in your revised version.

Beyond their comments, please verify similarities of parts of your manuscript with already published literature.

Reply. The MS has been thoroughly revised for grammar and punctuation and in order to avoid similarities with previously published literature. We have now rewritten and shortened many paragraphs and reorganized most of the sections as requested by the Reviewers.

I also have minor specific comments:

- Figure 5 has too extensive description, please insert it into the text.

Reply. We have now shortened the description of Figure 5 (Figure 4 in the revised version) by moving some sentences of the Figure caption to the main text.

- Specifically in the concluding remarks, I suggest include the meaning of abbreviation at first appearance (e.g. CT, SOC, GHG, and others), and subsequently use abbreviated form.

Reply. We checked the MS and included the meaning of abbreviations at the first appearance. We also followed your suggestion about including the meaning of abbreviation at first appearance in the concluding remarks.

Reviewer 1

It is an interesting reviewing work. The authors have presented many valuable things in this manuscript. However, they did not arrange the contents in good manner. Section 3 and 4 should be integrated in one section properly.

Reply. Thank you very much for your valuable comment. However, we do not agree with the suggestion made by the Reviewer regarding integrating Sections 3 and 4 into one single section for several reasons. First of all, because both sections are already very long at their present format, and thus we would have a very long Section 3 covering about 15 pages. Secondly, the content of both sections is different. While Section 3 is focused on describing the three principles of conservation agriculture, Section 4 is focused on Soil Processes, which fit perfectly with the second part of the MS title i.e., Principles, Processes, Practices and Policy Options. Nevertheless, to be clearer, we have now arranged the content as follows:  Principle (Section 3), Processes (Section 4), Practices (Section 5) and Policy options (Section 6). We have also renamed the Section 3 as “Principles: Conservation tillage, permanent plant cover and crop diversification” (i.e., the 3 principles of conservation agriculture) and re-arranged the subsections numbering in Section 3 in the revised version of the MS for an easier reading and understanding. Specifically, the three main subsections of Section 3 clearly refer to Conservation tillage (subsection 3.1), Permanent plant cover (subsection 3.2) and Crop diversification (subsection 3.3) in the revised version of the MS.

Section 3.1.1: Weed control is generally managed by herbicides or in some cases by crop rotation. Herbicides cause great damage to the soil system. I hope the author can consider carefully and add some biological control or mechanical control.

Reply. We agree with the Reviewer 1 comment on the detrimental impacts of herbicides in the soil system. Indeed, we had already raised this important issue in other sections of the review. Nevertheless, we have now made this point clearer in section 3.1. (section 3.1.1. in the previous version of the MS).

Section 3.3: The conversion from conventional tillage to CT increases SOC stocks worldwide. Add references please!

Reply. References were added as requested, references [2,27] in subsection 3.1.4. of the revised version of the MS.

Sections 3.8 and 3.9 should be integrated with sections 3.3 and 3.4.

Reply. We have rearranged the sections and subsections numbering (see the previous reply) to avoid misunderstandings. Indeed, this comment cannot be addressed because former sections 3.3 and 3.4 were referring to one of the principles of Conservation Agriculture (i.e., conservation tillage) while former sections 3.8 and 3.9 were referring to a different principle of Conservation Agriculture (i.e., permanent plant cover). Sections dealing with two different principles of conservation agriculture cannot be mixed. We believe that the new structuration of the subsections has improved the structure and clarity of the MS.  

Generally: The structure of section 3 and 4 should be rephrased.

Reply. As stated above (see the replies to the comments and suggestions made by the Editor), we have now rewritten, shortened and reorganized most of the MS sections.

Reviewer 2

The review is nicely complied covering all the aspects of conservation agriculture. Although it is a comprehensive review, during the reading I found that it may be reduced by avoiding very basic information and definitions.

Reply. Thank you for your appreciation. However, we prefer to maintain all definitions because the potential readers, for example, could have an expertise in conservation tillage but not in crop diversification or vice versa.

Older references (e.g. 1974) may either be deleted or replaced with the current citations.

Reply. Changed as requested, see updated reference [14].

Heading 2. Should be changed to ‘Adoption of conservation agriculture’

Reply. Changed as requested.

The conclusion section is very long and lengthy it should be shortened. The future prospect section should be incorporated before the conclusion section.

Reply. We have shortened the conclusion section focusing on which is really relevant. We have also changed the heading of the conclusion section to make more explicit and better encapsulate its contents, since apparently it was not clear.

I feel that all the tables and figures incorporated are adopted/taken from the already published literature. What is the contribution of authors in terms of Figures and Tables in this review?

Reply. The source was always reported in all tables and figures, and some of them are own elaborations of published data. However, and in agreement with the Reviewer comment, for the other figures that were not elaborated from us, we considered the following options during the revision.

1) We deleted or replaced those figures that were already published elsewhere and were not really necessary in this review, leaving the reference to the sources in the main text. This is the case of former figure 6 (section 5.2 of the revised version), and figures 10 to 13 of Section 6. Policy options that were replaced with an own elaboration based on the cited literature (figure 8 in the revised version).

2) We transformed the figures into tables. This is the case of one figure in section 3.13. of the original MS (please, note that by mistake it was indicated as Figure 1), that was transformed to table 4 (section 3.3.2. of the revised version).

Reviewer 3

The submitted review is based on a broad literature research. Conservation Agriculture has been known for decades, but only a minority of farmers use it, therefore this topic is still relevant.

I consider the submitted review to be of high quality and have only a few recommendations:

Authors should read the manuscript carefully and make minor corrections to typos (e.g. l. 90 - in in; l. 499 - .; l. 521 - mentiioned, l. 677 my/may, l.944 intotal....).

Reply. We checked and typos have been corrected here and in other parts of MS.

Abbreviations must always be explained - e.g. IPCC, PC, NPC...

Reply. We checked the MS and included the meaning of abbreviations at the first appearance (this was also requested by other Reviewers).

Figure 6 has too extensive description - insert it into the text.

Reply. This figure was deleted, but the cited literature was described in the main text.

Citations must always be in the same format - compare l. 207 and l. 209.

Reply. We followed the reference formatting style of the journal. In line 207 [44-45] indicates two consecutive references 44 and 45, in line 209 [37,40] indicates references 37 and 40 that are not consecutive.

Units must be in the same format - compare - l. 206 and l. 426.

Reply. We changed to “N2O emissions from soils” in line 206 because in this case there was no need to use the unit ha.

  1. 870 ...under three nitrogen fertilization levels (0, low, high) - What was the low/high dosage?

Reply. Fertilization levels were added as requested.

What is authors' opinion on the regenerative agriculture? Consider if this trend should be included in the review.

Reply. We prefer not including regenerative agriculture in the review. As reported in a recent review (Newton et al. 2020), no legal or regulatory definition of the term “regenerative agriculture” exists nor has a widely accepted definition emerged in common usage and may be helpful for individual users of the term “regenerative agriculture” to define it comprehensively for their own purpose and context.

Newton P, Civita N, Frankel-Goldwater L, Bartel K and Johns C (2020) What Is Regenerative Agriculture? A Review of Scholar and Practitioner Definitions Based on Processes and Outcomes. Front. Sustain. Food Syst. 4:577723. doi: 10.3389/fsufs.2020.577723

Reviewer 4

Manuscript soilsystems-2181992 “Conservation Agriculture and Soil Organic Carbon: Principles, Processes, Practices and Policy Options”

Authors discussed Conservation Agriculture and Soil Organic Carbon management as adoption strategies for sustainable agriculture. It is widely accepted that Conservation Agriculture is a holistic approach to cropping system management, intended for sustainable crop production without compromising environment.

Overall manuscript is well written; however, revision is required before the formal publication of manuscript.

General Comments

Minor English revision required to improve the manuscript and for common readers. Several place word replacements are required.

Reply. We checked the MS and improved the language style as much as possible.

Define abbreviation at first appearance, subsequently use abbreviated form e.g., SOC

Reply. We checked the MS and included the meaning of abbreviations at the first appearance. We also followed the Editor suggestion about including again the meaning of abbreviation at first appearance in the concluding remarks.

Figures used in the manuscript are directly added or adopted from already published articles, clearly mention at bottom of figure.

Reply. The source was always reported in all tables and figures, and some of them are own elaborations of published data. However, for the other figures that were not elaborated from us, we considered the following options during the revision.

1) We deleted or replaced those figures that were already published elsewhere and were not really necessary in this review, leaving the reference to the sources in the main text. This is the case of former figure 6 (section 5.2 of the revised version), and figures 10 to 13 of Section 6. Policy options that were replaced with an own elaboration based on the cited literature (figure 8 in the revised version).

2) We transformed the figures into tables. This is the case of one figure in section 3.13. of the original MS (please, note that by mistake it was indicated as Figure 1), that was transformed to table 4 (section 3.3.2. of the revised version).

Incorporate below mentioned important studies, if possible

Reply. See the specific comment on Suggested citations.

Conclusion is too long

Reply. We have changed the heading of the conclusion section to better capture its contents, address the comment to incorporate future prospects and shortened the length (as requested by Reviewer 2).

Specific comments

L-40: revise as “progressive soil degradation over time”

Reply. We rephrased as suggested.

L-43: better use “soil biota”

Reply. We rephrased as suggested.

L-87-88: Revised the text

Reply. Revised as requested.

L-142-151: Add suitable reference/s

Reply. Added as suggested.

L-200: Add “)”

Reply. Added as suggested.

L-208-209: Briefly add reasons of resilience against extreme rainfall events, droughts and warming events, to facilitate common readers

Reply. We have already explained the reasons why soils increase their resilience to extreme weather events as requested for improving readers understanding (lines 625-635 in the revised version of the MS)

L-212: revise the text

Reply. We have revised the text and made some re-phrasement for improving readers understanding.

L-219: replace “hamper” with “restrict”.

Reply. Replaced as suggested.

Suggested Citations

Reply. Among the citations suggested by the Reviewer, we included the following in the Policy section as references 166 and 167.

Panagos, P.; Montanarella, L.; Barbero, M.; Schneegans, A.; Aguglia, L.; Jones, A. Soil priorities in the European Union, Geoderma Reg. 2022, 29, e00510,

Panagos, P.; Van Liedekerke, M.; Borrelli, P.; Köninger, J.; Ballabio, C.;Orgiazzi, A.; Lugato, E.; Liakos, L.; Hervas, J.; Jones, A.;  Montanarella, L. European Soil Data Centre 2.0: Soil data and knowledge in support of the EU policies. Eur. J. Soil Sci. 2022, 73, e13315.

Panagos, P., M. Van Liedekerke, P. Borrelli, J. Köninger, C. Ballabio, A. Orgiazzi, E. Lugato, L. Liakos, J. Hervas, A. Jones and L. Montanarella, 2022: European Soil Data Centre 2.0: Soil data and knowledge in support of the EU policies. European Journal of Soil Science 73, e13315.

Köninger, J., P. Panagos, A. Jones, M. J. I. Briones and A. Orgiazzi, 2022: In defence of soil biodiversity: Towards an inclusive protection in the European Union. Biological Conservation 268, 109475.

Borrelli, P., P. Panagos, C. Alewell, C. Ballabio, H. de Oliveira Fagundes, N. Haregeweyn, E. Lugato, M. Maerker, J. Poesen, M. Vanmaercke and D. A. Robinson, 2023: Policy implications of multiple concurrent soil erosion processes in European farmland. Nature Sustainability 6, 103-112.

Panagos, P., L. Montanarella, M. Barbero, A. Schneegans, L. Aguglia and A. Jones, 2022: Soil priorities in the European Union. Geoderma Regional 29, e00510.

Borrelli, P., P. Panagos, C. Alewell, C. Ballabio, H. de Oliveira Fagundes, N. Haregeweyn, E. Lugato, M. Maerker, J. Poesen, M. Vanmaercke and D. A. Robinson, 2023: Policy implications of multiple concurrent soil erosion processes in European farmland. Nature Sustainability 6, 103-112.

Panagos, P., E. Lugato, C. Ballabio, I. Biavetti, L. Montanarella and P. Borrelli, 2022: Soil Erosion in Europe: From Policy Developments to Models, Indicators and New Research Challenges. In: R. Li, T. L. Napier, S. A. El-Swaify, M. Sabir and E. Rienzi eds. Global Degradation of Soil and Water Resources: Regional Assessment and Strategies. pp. 319-333. Springer Nature Singapore, Singapore.

Reviewer 2 Report

The review is nicely complied covering all the aspects of conservation agriculture. Although it is a comprehensive review, during the reading I found that it may be reduced by avoiding very basic information and definitions. Older references (e.g. 1974) may either be deleted or replaced with the current citations.

Heading 2. Should be changed to ‘Adoption of conservation agriculture’

The conclusion section is very long and lengthy it should be shortened.

The future prospect section should be incorporated before the conclusion section.

I feel that all the tables and figures incorporated are adopted/taken from the already published literature. What is the contribution of authors in terms of Figures and Tables in this review? 

Academic Editor Notes

Dear authors,

Thank you for submitting your manuscript to Soil Systems.

Based on the reviewers’ comments, my decision is accept the manuscript after major revision.

Four reviewers revised your manuscript, and I recommend consider their comments and suggestions in your revised version.

Beyond their comments, please verify similarities of parts of your manuscript with already published literature.

Reply. The MS has been thoroughly revised for grammar and punctuation and in order to avoid similarities with previously published literature. We have now rewritten and shortened many paragraphs and reorganized most of the sections as requested by the Reviewers.

I also have minor specific comments:

- Figure 5 has too extensive description, please insert it into the text.

Reply. We have now shortened the description of Figure 5 (Figure 4 in the revised version) by moving some sentences of the Figure caption to the main text.

- Specifically in the concluding remarks, I suggest include the meaning of abbreviation at first appearance (e.g. CT, SOC, GHG, and others), and subsequently use abbreviated form.

Reply. We checked the MS and included the meaning of abbreviations at the first appearance. We also followed your suggestion about including the meaning of abbreviation at first appearance in the concluding remarks.

Reviewer 1

It is an interesting reviewing work. The authors have presented many valuable things in this manuscript. However, they did not arrange the contents in good manner. Section 3 and 4 should be integrated in one section properly.

Reply. Thank you very much for your valuable comment. However, we do not agree with the suggestion made by the Reviewer regarding integrating Sections 3 and 4 into one single section for several reasons. First of all, because both sections are already very long at their present format, and thus we would have a very long Section 3 covering about 15 pages. Secondly, the content of both sections is different. While Section 3 is focused on describing the three principles of conservation agriculture, Section 4 is focused on Soil Processes, which fit perfectly with the second part of the MS title i.e., Principles, Processes, Practices and Policy Options. Nevertheless, to be clearer, we have now arranged the content as follows:  Principle (Section 3), Processes (Section 4), Practices (Section 5) and Policy options (Section 6). We have also renamed the Section 3 as “Principles: Conservation tillage, permanent plant cover and crop diversification” (i.e., the 3 principles of conservation agriculture) and re-arranged the subsections numbering in Section 3 in the revised version of the MS for an easier reading and understanding. Specifically, the three main subsections of Section 3 clearly refer to Conservation tillage (subsection 3.1), Permanent plant cover (subsection 3.2) and Crop diversification (subsection 3.3) in the revised version of the MS.

Section 3.1.1: Weed control is generally managed by herbicides or in some cases by crop rotation. Herbicides cause great damage to the soil system. I hope the author can consider carefully and add some biological control or mechanical control.

Reply. We agree with the Reviewer 1 comment on the detrimental impacts of herbicides in the soil system. Indeed, we had already raised this important issue in other sections of the review. Nevertheless, we have now made this point clearer in section 3.1. (section 3.1.1. in the previous version of the MS).

Section 3.3: The conversion from conventional tillage to CT increases SOC stocks worldwide. Add references please!

Reply. References were added as requested, references [2,27] in subsection 3.1.4. of the revised version of the MS.

Sections 3.8 and 3.9 should be integrated with sections 3.3 and 3.4.

Reply. We have rearranged the sections and subsections numbering (see the previous reply) to avoid misunderstandings. Indeed, this comment cannot be addressed because former sections 3.3 and 3.4 were referring to one of the principles of Conservation Agriculture (i.e., conservation tillage) while former sections 3.8 and 3.9 were referring to a different principle of Conservation Agriculture (i.e., permanent plant cover). Sections dealing with two different principles of conservation agriculture cannot be mixed. We believe that the new structuration of the subsections has improved the structure and clarity of the MS.  

Generally: The structure of section 3 and 4 should be rephrased.

Reply. As stated above (see the replies to the comments and suggestions made by the Editor), we have now rewritten, shortened and reorganized most of the MS sections.

Reviewer 2

The review is nicely complied covering all the aspects of conservation agriculture. Although it is a comprehensive review, during the reading I found that it may be reduced by avoiding very basic information and definitions.

Reply. Thank you for your appreciation. However, we prefer to maintain all definitions because the potential readers, for example, could have an expertise in conservation tillage but not in crop diversification or vice versa.

Older references (e.g. 1974) may either be deleted or replaced with the current citations.

Reply. Changed as requested, see updated reference [14].

Heading 2. Should be changed to ‘Adoption of conservation agriculture’

Reply. Changed as requested.

The conclusion section is very long and lengthy it should be shortened. The future prospect section should be incorporated before the conclusion section.

Reply. We have shortened the conclusion section focusing on which is really relevant. We have also changed the heading of the conclusion section to make more explicit and better encapsulate its contents, since apparently it was not clear.

I feel that all the tables and figures incorporated are adopted/taken from the already published literature. What is the contribution of authors in terms of Figures and Tables in this review?

Reply. The source was always reported in all tables and figures, and some of them are own elaborations of published data. However, and in agreement with the Reviewer comment, for the other figures that were not elaborated from us, we considered the following options during the revision.

1) We deleted or replaced those figures that were already published elsewhere and were not really necessary in this review, leaving the reference to the sources in the main text. This is the case of former figure 6 (section 5.2 of the revised version), and figures 10 to 13 of Section 6. Policy options that were replaced with an own elaboration based on the cited literature (figure 8 in the revised version).

2) We transformed the figures into tables. This is the case of one figure in section 3.13. of the original MS (please, note that by mistake it was indicated as Figure 1), that was transformed to table 4 (section 3.3.2. of the revised version).

Reviewer 3

The submitted review is based on a broad literature research. Conservation Agriculture has been known for decades, but only a minority of farmers use it, therefore this topic is still relevant.

I consider the submitted review to be of high quality and have only a few recommendations:

Authors should read the manuscript carefully and make minor corrections to typos (e.g. l. 90 - in in; l. 499 - .; l. 521 - mentiioned, l. 677 my/may, l.944 intotal....).

Reply. We checked and typos have been corrected here and in other parts of MS.

Abbreviations must always be explained - e.g. IPCC, PC, NPC...

Reply. We checked the MS and included the meaning of abbreviations at the first appearance (this was also requested by other Reviewers).

Figure 6 has too extensive description - insert it into the text.

Reply. This figure was deleted, but the cited literature was described in the main text.

Citations must always be in the same format - compare l. 207 and l. 209.

Reply. We followed the reference formatting style of the journal. In line 207 [44-45] indicates two consecutive references 44 and 45, in line 209 [37,40] indicates references 37 and 40 that are not consecutive.

Units must be in the same format - compare - l. 206 and l. 426.

Reply. We changed to “N2O emissions from soils” in line 206 because in this case there was no need to use the unit ha.

  1. 870 ...under three nitrogen fertilization levels (0, low, high) - What was the low/high dosage?

Reply. Fertilization levels were added as requested.

What is authors' opinion on the regenerative agriculture? Consider if this trend should be included in the review.

Reply. We prefer not including regenerative agriculture in the review. As reported in a recent review (Newton et al. 2020), no legal or regulatory definition of the term “regenerative agriculture” exists nor has a widely accepted definition emerged in common usage and may be helpful for individual users of the term “regenerative agriculture” to define it comprehensively for their own purpose and context.

Newton P, Civita N, Frankel-Goldwater L, Bartel K and Johns C (2020) What Is Regenerative Agriculture? A Review of Scholar and Practitioner Definitions Based on Processes and Outcomes. Front. Sustain. Food Syst. 4:577723. doi: 10.3389/fsufs.2020.577723

Reviewer 4

Manuscript soilsystems-2181992 “Conservation Agriculture and Soil Organic Carbon: Principles, Processes, Practices and Policy Options”

Authors discussed Conservation Agriculture and Soil Organic Carbon management as adoption strategies for sustainable agriculture. It is widely accepted that Conservation Agriculture is a holistic approach to cropping system management, intended for sustainable crop production without compromising environment.

Overall manuscript is well written; however, revision is required before the formal publication of manuscript.

General Comments

Minor English revision required to improve the manuscript and for common readers. Several place word replacements are required.

Reply. We checked the MS and improved the language style as much as possible.

Define abbreviation at first appearance, subsequently use abbreviated form e.g., SOC

Reply. We checked the MS and included the meaning of abbreviations at the first appearance. We also followed the Editor suggestion about including again the meaning of abbreviation at first appearance in the concluding remarks.

Figures used in the manuscript are directly added or adopted from already published articles, clearly mention at bottom of figure.

Reply. The source was always reported in all tables and figures, and some of them are own elaborations of published data. However, for the other figures that were not elaborated from us, we considered the following options during the revision.

1) We deleted or replaced those figures that were already published elsewhere and were not really necessary in this review, leaving the reference to the sources in the main text. This is the case of former figure 6 (section 5.2 of the revised version), and figures 10 to 13 of Section 6. Policy options that were replaced with an own elaboration based on the cited literature (figure 8 in the revised version).

2) We transformed the figures into tables. This is the case of one figure in section 3.13. of the original MS (please, note that by mistake it was indicated as Figure 1), that was transformed to table 4 (section 3.3.2. of the revised version).

Incorporate below mentioned important studies, if possible

Reply. See the specific comment on Suggested citations.

Conclusion is too long

Reply. We have changed the heading of the conclusion section to better capture its contents, address the comment to incorporate future prospects and shortened the length (as requested by Reviewer 2).

Specific comments

L-40: revise as “progressive soil degradation over time”

Reply. We rephrased as suggested.

L-43: better use “soil biota”

Reply. We rephrased as suggested.

L-87-88: Revised the text

Reply. Revised as requested.

L-142-151: Add suitable reference/s

Reply. Added as suggested.

L-200: Add “)”

Reply. Added as suggested.

L-208-209: Briefly add reasons of resilience against extreme rainfall events, droughts and warming events, to facilitate common readers

Reply. We have already explained the reasons why soils increase their resilience to extreme weather events as requested for improving readers understanding (lines 625-635 in the revised version of the MS)

L-212: revise the text

Reply. We have revised the text and made some re-phrasement for improving readers understanding.

L-219: replace “hamper” with “restrict”.

Reply. Replaced as suggested.

Suggested Citations

Reply. Among the citations suggested by the Reviewer, we included the following in the Policy section as references 166 and 167.

Panagos, P.; Montanarella, L.; Barbero, M.; Schneegans, A.; Aguglia, L.; Jones, A. Soil priorities in the European Union, Geoderma Reg. 2022, 29, e00510,

Panagos, P.; Van Liedekerke, M.; Borrelli, P.; Köninger, J.; Ballabio, C.;Orgiazzi, A.; Lugato, E.; Liakos, L.; Hervas, J.; Jones, A.;  Montanarella, L. European Soil Data Centre 2.0: Soil data and knowledge in support of the EU policies. Eur. J. Soil Sci. 2022, 73, e13315.

Panagos, P., M. Van Liedekerke, P. Borrelli, J. Köninger, C. Ballabio, A. Orgiazzi, E. Lugato, L. Liakos, J. Hervas, A. Jones and L. Montanarella, 2022: European Soil Data Centre 2.0: Soil data and knowledge in support of the EU policies. European Journal of Soil Science 73, e13315.

Köninger, J., P. Panagos, A. Jones, M. J. I. Briones and A. Orgiazzi, 2022: In defence of soil biodiversity: Towards an inclusive protection in the European Union. Biological Conservation 268, 109475.

Borrelli, P., P. Panagos, C. Alewell, C. Ballabio, H. de Oliveira Fagundes, N. Haregeweyn, E. Lugato, M. Maerker, J. Poesen, M. Vanmaercke and D. A. Robinson, 2023: Policy implications of multiple concurrent soil erosion processes in European farmland. Nature Sustainability 6, 103-112.

Panagos, P., L. Montanarella, M. Barbero, A. Schneegans, L. Aguglia and A. Jones, 2022: Soil priorities in the European Union. Geoderma Regional 29, e00510.

Borrelli, P., P. Panagos, C. Alewell, C. Ballabio, H. de Oliveira Fagundes, N. Haregeweyn, E. Lugato, M. Maerker, J. Poesen, M. Vanmaercke and D. A. Robinson, 2023: Policy implications of multiple concurrent soil erosion processes in European farmland. Nature Sustainability 6, 103-112.

Panagos, P., E. Lugato, C. Ballabio, I. Biavetti, L. Montanarella and P. Borrelli, 2022: Soil Erosion in Europe: From Policy Developments to Models, Indicators and New Research Challenges. In: R. Li, T. L. Napier, S. A. El-Swaify, M. Sabir and E. Rienzi eds. Global Degradation of Soil and Water Resources: Regional Assessment and Strategies. pp. 319-333. Springer Nature Singapore, Singapore.

Reviewer 3 Report

The submitted review is based on a broad literature research. Conservation Agriculture has been known for decades, but only a minority of farmers use it, therefore this topic is still relevant.

I consider the submitted review to be of high quality and have only a few recommendations:

Authors should read the manuscript carefully and make minor corrections to typos (e.g. l. 90 - in in; l. 499 - .; l. 521 - mentiioned, l. 677 my/may, l.944 intotal....).

Abbreviations must always be explained - e.g. IPCC, PC, NPC...

Figure 6 has too extensive description - insert it into the text.

Citations must always be in the same format - compare l. 207

 and l. 209.

Units must be in the same format - compare - l. 206 and l. 426.

l. 870 ...under three nitrogen fertilization levels (0, low, high) - What was the low/high dosage?

What is authors' opinion on the regenerative agriculture? Consider if this trend should be included in the review.

Academic Editor Notes

Dear authors,

Thank you for submitting your manuscript to Soil Systems.

Based on the reviewers’ comments, my decision is accept the manuscript after major revision.

Four reviewers revised your manuscript, and I recommend consider their comments and suggestions in your revised version.

Beyond their comments, please verify similarities of parts of your manuscript with already published literature.

Reply. The MS has been thoroughly revised for grammar and punctuation and in order to avoid similarities with previously published literature. We have now rewritten and shortened many paragraphs and reorganized most of the sections as requested by the Reviewers.

I also have minor specific comments:

- Figure 5 has too extensive description, please insert it into the text.

Reply. We have now shortened the description of Figure 5 (Figure 4 in the revised version) by moving some sentences of the Figure caption to the main text.

- Specifically in the concluding remarks, I suggest include the meaning of abbreviation at first appearance (e.g. CT, SOC, GHG, and others), and subsequently use abbreviated form.

Reply. We checked the MS and included the meaning of abbreviations at the first appearance. We also followed your suggestion about including the meaning of abbreviation at first appearance in the concluding remarks.

Reviewer 1

It is an interesting reviewing work. The authors have presented many valuable things in this manuscript. However, they did not arrange the contents in good manner. Section 3 and 4 should be integrated in one section properly.

Reply. Thank you very much for your valuable comment. However, we do not agree with the suggestion made by the Reviewer regarding integrating Sections 3 and 4 into one single section for several reasons. First of all, because both sections are already very long at their present format, and thus we would have a very long Section 3 covering about 15 pages. Secondly, the content of both sections is different. While Section 3 is focused on describing the three principles of conservation agriculture, Section 4 is focused on Soil Processes, which fit perfectly with the second part of the MS title i.e., Principles, Processes, Practices and Policy Options. Nevertheless, to be clearer, we have now arranged the content as follows:  Principle (Section 3), Processes (Section 4), Practices (Section 5) and Policy options (Section 6). We have also renamed the Section 3 as “Principles: Conservation tillage, permanent plant cover and crop diversification” (i.e., the 3 principles of conservation agriculture) and re-arranged the subsections numbering in Section 3 in the revised version of the MS for an easier reading and understanding. Specifically, the three main subsections of Section 3 clearly refer to Conservation tillage (subsection 3.1), Permanent plant cover (subsection 3.2) and Crop diversification (subsection 3.3) in the revised version of the MS.

Section 3.1.1: Weed control is generally managed by herbicides or in some cases by crop rotation. Herbicides cause great damage to the soil system. I hope the author can consider carefully and add some biological control or mechanical control.

Reply. We agree with the Reviewer 1 comment on the detrimental impacts of herbicides in the soil system. Indeed, we had already raised this important issue in other sections of the review. Nevertheless, we have now made this point clearer in section 3.1. (section 3.1.1. in the previous version of the MS).

Section 3.3: The conversion from conventional tillage to CT increases SOC stocks worldwide. Add references please!

Reply. References were added as requested, references [2,27] in subsection 3.1.4. of the revised version of the MS.

Sections 3.8 and 3.9 should be integrated with sections 3.3 and 3.4.

Reply. We have rearranged the sections and subsections numbering (see the previous reply) to avoid misunderstandings. Indeed, this comment cannot be addressed because former sections 3.3 and 3.4 were referring to one of the principles of Conservation Agriculture (i.e., conservation tillage) while former sections 3.8 and 3.9 were referring to a different principle of Conservation Agriculture (i.e., permanent plant cover). Sections dealing with two different principles of conservation agriculture cannot be mixed. We believe that the new structuration of the subsections has improved the structure and clarity of the MS.  

Generally: The structure of section 3 and 4 should be rephrased.

Reply. As stated above (see the replies to the comments and suggestions made by the Editor), we have now rewritten, shortened and reorganized most of the MS sections.

Reviewer 2

The review is nicely complied covering all the aspects of conservation agriculture. Although it is a comprehensive review, during the reading I found that it may be reduced by avoiding very basic information and definitions.

Reply. Thank you for your appreciation. However, we prefer to maintain all definitions because the potential readers, for example, could have an expertise in conservation tillage but not in crop diversification or vice versa.

Older references (e.g. 1974) may either be deleted or replaced with the current citations.

Reply. Changed as requested, see updated reference [14].

Heading 2. Should be changed to ‘Adoption of conservation agriculture’

Reply. Changed as requested.

The conclusion section is very long and lengthy it should be shortened. The future prospect section should be incorporated before the conclusion section.

Reply. We have shortened the conclusion section focusing on which is really relevant. We have also changed the heading of the conclusion section to make more explicit and better encapsulate its contents, since apparently it was not clear.

I feel that all the tables and figures incorporated are adopted/taken from the already published literature. What is the contribution of authors in terms of Figures and Tables in this review?

Reply. The source was always reported in all tables and figures, and some of them are own elaborations of published data. However, and in agreement with the Reviewer comment, for the other figures that were not elaborated from us, we considered the following options during the revision.

1) We deleted or replaced those figures that were already published elsewhere and were not really necessary in this review, leaving the reference to the sources in the main text. This is the case of former figure 6 (section 5.2 of the revised version), and figures 10 to 13 of Section 6. Policy options that were replaced with an own elaboration based on the cited literature (figure 8 in the revised version).

2) We transformed the figures into tables. This is the case of one figure in section 3.13. of the original MS (please, note that by mistake it was indicated as Figure 1), that was transformed to table 4 (section 3.3.2. of the revised version).

Reviewer 3

The submitted review is based on a broad literature research. Conservation Agriculture has been known for decades, but only a minority of farmers use it, therefore this topic is still relevant.

I consider the submitted review to be of high quality and have only a few recommendations:

Authors should read the manuscript carefully and make minor corrections to typos (e.g. l. 90 - in in; l. 499 - .; l. 521 - mentiioned, l. 677 my/may, l.944 intotal....).

Reply. We checked and typos have been corrected here and in other parts of MS.

Abbreviations must always be explained - e.g. IPCC, PC, NPC...

Reply. We checked the MS and included the meaning of abbreviations at the first appearance (this was also requested by other Reviewers).

Figure 6 has too extensive description - insert it into the text.

Reply. This figure was deleted, but the cited literature was described in the main text.

Citations must always be in the same format - compare l. 207 and l. 209.

Reply. We followed the reference formatting style of the journal. In line 207 [44-45] indicates two consecutive references 44 and 45, in line 209 [37,40] indicates references 37 and 40 that are not consecutive.

Units must be in the same format - compare - l. 206 and l. 426.

Reply. We changed to “N2O emissions from soils” in line 206 because in this case there was no need to use the unit ha.

  1. 870 ...under three nitrogen fertilization levels (0, low, high) - What was the low/high dosage?

Reply. Fertilization levels were added as requested.

What is authors' opinion on the regenerative agriculture? Consider if this trend should be included in the review.

Reply. We prefer not including regenerative agriculture in the review. As reported in a recent review (Newton et al. 2020), no legal or regulatory definition of the term “regenerative agriculture” exists nor has a widely accepted definition emerged in common usage and may be helpful for individual users of the term “regenerative agriculture” to define it comprehensively for their own purpose and context.

Newton P, Civita N, Frankel-Goldwater L, Bartel K and Johns C (2020) What Is Regenerative Agriculture? A Review of Scholar and Practitioner Definitions Based on Processes and Outcomes. Front. Sustain. Food Syst. 4:577723. doi: 10.3389/fsufs.2020.577723

Reviewer 4

Manuscript soilsystems-2181992 “Conservation Agriculture and Soil Organic Carbon: Principles, Processes, Practices and Policy Options”

Authors discussed Conservation Agriculture and Soil Organic Carbon management as adoption strategies for sustainable agriculture. It is widely accepted that Conservation Agriculture is a holistic approach to cropping system management, intended for sustainable crop production without compromising environment.

Overall manuscript is well written; however, revision is required before the formal publication of manuscript.

General Comments

Minor English revision required to improve the manuscript and for common readers. Several place word replacements are required.

Reply. We checked the MS and improved the language style as much as possible.

Define abbreviation at first appearance, subsequently use abbreviated form e.g., SOC

Reply. We checked the MS and included the meaning of abbreviations at the first appearance. We also followed the Editor suggestion about including again the meaning of abbreviation at first appearance in the concluding remarks.

Figures used in the manuscript are directly added or adopted from already published articles, clearly mention at bottom of figure.

Reply. The source was always reported in all tables and figures, and some of them are own elaborations of published data. However, for the other figures that were not elaborated from us, we considered the following options during the revision.

1) We deleted or replaced those figures that were already published elsewhere and were not really necessary in this review, leaving the reference to the sources in the main text. This is the case of former figure 6 (section 5.2 of the revised version), and figures 10 to 13 of Section 6. Policy options that were replaced with an own elaboration based on the cited literature (figure 8 in the revised version).

2) We transformed the figures into tables. This is the case of one figure in section 3.13. of the original MS (please, note that by mistake it was indicated as Figure 1), that was transformed to table 4 (section 3.3.2. of the revised version).

Incorporate below mentioned important studies, if possible

Reply. See the specific comment on Suggested citations.

Conclusion is too long

Reply. We have changed the heading of the conclusion section to better capture its contents, address the comment to incorporate future prospects and shortened the length (as requested by Reviewer 2).

Specific comments

L-40: revise as “progressive soil degradation over time”

Reply. We rephrased as suggested.

L-43: better use “soil biota”

Reply. We rephrased as suggested.

L-87-88: Revised the text

Reply. Revised as requested.

L-142-151: Add suitable reference/s

Reply. Added as suggested.

L-200: Add “)”

Reply. Added as suggested.

L-208-209: Briefly add reasons of resilience against extreme rainfall events, droughts and warming events, to facilitate common readers

Reply. We have already explained the reasons why soils increase their resilience to extreme weather events as requested for improving readers understanding (lines 625-635 in the revised version of the MS)

L-212: revise the text

Reply. We have revised the text and made some re-phrasement for improving readers understanding.

L-219: replace “hamper” with “restrict”.

Reply. Replaced as suggested.

Suggested Citations

Reply. Among the citations suggested by the Reviewer, we included the following in the Policy section as references 166 and 167.

Panagos, P.; Montanarella, L.; Barbero, M.; Schneegans, A.; Aguglia, L.; Jones, A. Soil priorities in the European Union, Geoderma Reg. 2022, 29, e00510,

Panagos, P.; Van Liedekerke, M.; Borrelli, P.; Köninger, J.; Ballabio, C.;Orgiazzi, A.; Lugato, E.; Liakos, L.; Hervas, J.; Jones, A.;  Montanarella, L. European Soil Data Centre 2.0: Soil data and knowledge in support of the EU policies. Eur. J. Soil Sci. 2022, 73, e13315.

Panagos, P., M. Van Liedekerke, P. Borrelli, J. Köninger, C. Ballabio, A. Orgiazzi, E. Lugato, L. Liakos, J. Hervas, A. Jones and L. Montanarella, 2022: European Soil Data Centre 2.0: Soil data and knowledge in support of the EU policies. European Journal of Soil Science 73, e13315.

Köninger, J., P. Panagos, A. Jones, M. J. I. Briones and A. Orgiazzi, 2022: In defence of soil biodiversity: Towards an inclusive protection in the European Union. Biological Conservation 268, 109475.

Borrelli, P., P. Panagos, C. Alewell, C. Ballabio, H. de Oliveira Fagundes, N. Haregeweyn, E. Lugato, M. Maerker, J. Poesen, M. Vanmaercke and D. A. Robinson, 2023: Policy implications of multiple concurrent soil erosion processes in European farmland. Nature Sustainability 6, 103-112.

Panagos, P., L. Montanarella, M. Barbero, A. Schneegans, L. Aguglia and A. Jones, 2022: Soil priorities in the European Union. Geoderma Regional 29, e00510.

Borrelli, P., P. Panagos, C. Alewell, C. Ballabio, H. de Oliveira Fagundes, N. Haregeweyn, E. Lugato, M. Maerker, J. Poesen, M. Vanmaercke and D. A. Robinson, 2023: Policy implications of multiple concurrent soil erosion processes in European farmland. Nature Sustainability 6, 103-112.

Panagos, P., E. Lugato, C. Ballabio, I. Biavetti, L. Montanarella and P. Borrelli, 2022: Soil Erosion in Europe: From Policy Developments to Models, Indicators and New Research Challenges. In: R. Li, T. L. Napier, S. A. El-Swaify, M. Sabir and E. Rienzi eds. Global Degradation of Soil and Water Resources: Regional Assessment and Strategies. pp. 319-333. Springer Nature Singapore, Singapore.

Reviewer 4 Report

Manuscript soilsystems-2181992 Conservation Agriculture and Soil Organic Carbon: Principles, Processes, Practices and Policy Options

Authors discussed Conservation Agriculture and Soil Organic Carbon management as adoption strategies for sustainable agriculture. It is widely accepted that Conservation Agriculture is a holistic approach to cropping system management, intended for sustainable crop production without compromising environment.

Overall manuscript is well written, however, revision is required before the formal publication of manuscript.

General Comments

·       Minor English revision required to improve the manuscript and for common readers. Several place word replacements are required

·       Define abbreviation at first appearance, subsequently use abbreviated form e.g., SOC

·       Figures used in the manuscript are directly added or adopted from already published articles, clearly mention at bottom of figure.

·       Incorporate below mentioned important studies, if possible

·       Conclusion is too long

Specific comments

·       L-40: revise as “progressive soil degradation over time”

·       L-43: better use “soil biota”

·       L-87-88: Revised the text

·       L-142-151: Add suitable reference/s

·       L-200: Add “)”

·       L-208-209: Briefly add reasons of resilience against extreme rainfall events, droughts and warming events, to facilitate common readers

·       L-212: revise the text

·       L-219: replace “hamper” with “restrict”

Suggested Citations

·       Panagos, P., M. Van Liedekerke, P. Borrelli, J. Köninger, C. Ballabio, A. Orgiazzi, E. Lugato, L. Liakos, J. Hervas, A. Jones and L. Montanarella, 2022: European Soil Data Centre 2.0: Soil data and knowledge in support of the EU policies. European Journal of Soil Science 73, e13315.

·       Köninger, J., P. Panagos, A. Jones, M. J. I. Briones and A. Orgiazzi, 2022: In defence of soil biodiversity: Towards an inclusive protection in the European Union. Biological Conservation 268, 109475.

·       Borrelli, P., P. Panagos, C. Alewell, C. Ballabio, H. de Oliveira Fagundes, N. Haregeweyn, E. Lugato, M. Maerker, J. Poesen, M. Vanmaercke and D. A. Robinson, 2023: Policy implications of multiple concurrent soil erosion processes in European farmland. Nature Sustainability 6, 103-112.

·       Panagos, P., L. Montanarella, M. Barbero, A. Schneegans, L. Aguglia and A. Jones, 2022: Soil priorities in the European Union. Geoderma Regional 29, e00510.

·       Borrelli, P., P. Panagos, C. Alewell, C. Ballabio, H. de Oliveira Fagundes, N. Haregeweyn, E. Lugato, M. Maerker, J. Poesen, M. Vanmaercke and D. A. Robinson, 2023: Policy implications of multiple concurrent soil erosion processes in European farmland. Nature Sustainability 6, 103-112.

·       Panagos, P., E. Lugato, C. Ballabio, I. Biavetti, L. Montanarella and P. Borrelli, 2022: Soil Erosion in Europe: From Policy Developments to Models, Indicators and New Research Challenges. In: R. Li, T. L. Napier, S. A. El-Swaify, M. Sabir and E. Rienzi eds. Global Degradation of Soil and Water Resources: Regional Assessment and Strategies. pp. 319-333. Springer Nature Singapore, Singapore.

Academic Editor Notes

Dear authors,

Thank you for submitting your manuscript to Soil Systems.

Based on the reviewers’ comments, my decision is accept the manuscript after major revision.

Four reviewers revised your manuscript, and I recommend consider their comments and suggestions in your revised version.

Beyond their comments, please verify similarities of parts of your manuscript with already published literature.

Reply. The MS has been thoroughly revised for grammar and punctuation and in order to avoid similarities with previously published literature. We have now rewritten and shortened many paragraphs and reorganized most of the sections as requested by the Reviewers.

I also have minor specific comments:

- Figure 5 has too extensive description, please insert it into the text.

Reply. We have now shortened the description of Figure 5 (Figure 4 in the revised version) by moving some sentences of the Figure caption to the main text.

- Specifically in the concluding remarks, I suggest include the meaning of abbreviation at first appearance (e.g. CT, SOC, GHG, and others), and subsequently use abbreviated form.

Reply. We checked the MS and included the meaning of abbreviations at the first appearance. We also followed your suggestion about including the meaning of abbreviation at first appearance in the concluding remarks.

Reviewer 1

It is an interesting reviewing work. The authors have presented many valuable things in this manuscript. However, they did not arrange the contents in good manner. Section 3 and 4 should be integrated in one section properly.

Reply. Thank you very much for your valuable comment. However, we do not agree with the suggestion made by the Reviewer regarding integrating Sections 3 and 4 into one single section for several reasons. First of all, because both sections are already very long at their present format, and thus we would have a very long Section 3 covering about 15 pages. Secondly, the content of both sections is different. While Section 3 is focused on describing the three principles of conservation agriculture, Section 4 is focused on Soil Processes, which fit perfectly with the second part of the MS title i.e., Principles, Processes, Practices and Policy Options. Nevertheless, to be clearer, we have now arranged the content as follows:  Principle (Section 3), Processes (Section 4), Practices (Section 5) and Policy options (Section 6). We have also renamed the Section 3 as “Principles: Conservation tillage, permanent plant cover and crop diversification” (i.e., the 3 principles of conservation agriculture) and re-arranged the subsections numbering in Section 3 in the revised version of the MS for an easier reading and understanding. Specifically, the three main subsections of Section 3 clearly refer to Conservation tillage (subsection 3.1), Permanent plant cover (subsection 3.2) and Crop diversification (subsection 3.3) in the revised version of the MS.

Section 3.1.1: Weed control is generally managed by herbicides or in some cases by crop rotation. Herbicides cause great damage to the soil system. I hope the author can consider carefully and add some biological control or mechanical control.

Reply. We agree with the Reviewer 1 comment on the detrimental impacts of herbicides in the soil system. Indeed, we had already raised this important issue in other sections of the review. Nevertheless, we have now made this point clearer in section 3.1. (section 3.1.1. in the previous version of the MS).

Section 3.3: The conversion from conventional tillage to CT increases SOC stocks worldwide. Add references please!

Reply. References were added as requested, references [2,27] in subsection 3.1.4. of the revised version of the MS.

Sections 3.8 and 3.9 should be integrated with sections 3.3 and 3.4.

Reply. We have rearranged the sections and subsections numbering (see the previous reply) to avoid misunderstandings. Indeed, this comment cannot be addressed because former sections 3.3 and 3.4 were referring to one of the principles of Conservation Agriculture (i.e., conservation tillage) while former sections 3.8 and 3.9 were referring to a different principle of Conservation Agriculture (i.e., permanent plant cover). Sections dealing with two different principles of conservation agriculture cannot be mixed. We believe that the new structuration of the subsections has improved the structure and clarity of the MS.  

Generally: The structure of section 3 and 4 should be rephrased.

Reply. As stated above (see the replies to the comments and suggestions made by the Editor), we have now rewritten, shortened and reorganized most of the MS sections.

Reviewer 2

The review is nicely complied covering all the aspects of conservation agriculture. Although it is a comprehensive review, during the reading I found that it may be reduced by avoiding very basic information and definitions.

Reply. Thank you for your appreciation. However, we prefer to maintain all definitions because the potential readers, for example, could have an expertise in conservation tillage but not in crop diversification or vice versa.

Older references (e.g. 1974) may either be deleted or replaced with the current citations.

Reply. Changed as requested, see updated reference [14].

Heading 2. Should be changed to ‘Adoption of conservation agriculture’

Reply. Changed as requested.

The conclusion section is very long and lengthy it should be shortened. The future prospect section should be incorporated before the conclusion section.

Reply. We have shortened the conclusion section focusing on which is really relevant. We have also changed the heading of the conclusion section to make more explicit and better encapsulate its contents, since apparently it was not clear.

I feel that all the tables and figures incorporated are adopted/taken from the already published literature. What is the contribution of authors in terms of Figures and Tables in this review?

Reply. The source was always reported in all tables and figures, and some of them are own elaborations of published data. However, and in agreement with the Reviewer comment, for the other figures that were not elaborated from us, we considered the following options during the revision.

1) We deleted or replaced those figures that were already published elsewhere and were not really necessary in this review, leaving the reference to the sources in the main text. This is the case of former figure 6 (section 5.2 of the revised version), and figures 10 to 13 of Section 6. Policy options that were replaced with an own elaboration based on the cited literature (figure 8 in the revised version).

2) We transformed the figures into tables. This is the case of one figure in section 3.13. of the original MS (please, note that by mistake it was indicated as Figure 1), that was transformed to table 4 (section 3.3.2. of the revised version).

Reviewer 3

The submitted review is based on a broad literature research. Conservation Agriculture has been known for decades, but only a minority of farmers use it, therefore this topic is still relevant.

I consider the submitted review to be of high quality and have only a few recommendations:

Authors should read the manuscript carefully and make minor corrections to typos (e.g. l. 90 - in in; l. 499 - .; l. 521 - mentiioned, l. 677 my/may, l.944 intotal....).

Reply. We checked and typos have been corrected here and in other parts of MS.

Abbreviations must always be explained - e.g. IPCC, PC, NPC...

Reply. We checked the MS and included the meaning of abbreviations at the first appearance (this was also requested by other Reviewers).

Figure 6 has too extensive description - insert it into the text.

Reply. This figure was deleted, but the cited literature was described in the main text.

Citations must always be in the same format - compare l. 207 and l. 209.

Reply. We followed the reference formatting style of the journal. In line 207 [44-45] indicates two consecutive references 44 and 45, in line 209 [37,40] indicates references 37 and 40 that are not consecutive.

Units must be in the same format - compare - l. 206 and l. 426.

Reply. We changed to “N2O emissions from soils” in line 206 because in this case there was no need to use the unit ha.

  1. 870 ...under three nitrogen fertilization levels (0, low, high) - What was the low/high dosage?

Reply. Fertilization levels were added as requested.

What is authors' opinion on the regenerative agriculture? Consider if this trend should be included in the review.

Reply. We prefer not including regenerative agriculture in the review. As reported in a recent review (Newton et al. 2020), no legal or regulatory definition of the term “regenerative agriculture” exists nor has a widely accepted definition emerged in common usage and may be helpful for individual users of the term “regenerative agriculture” to define it comprehensively for their own purpose and context.

Newton P, Civita N, Frankel-Goldwater L, Bartel K and Johns C (2020) What Is Regenerative Agriculture? A Review of Scholar and Practitioner Definitions Based on Processes and Outcomes. Front. Sustain. Food Syst. 4:577723. doi: 10.3389/fsufs.2020.577723

Reviewer 4

Manuscript soilsystems-2181992 “Conservation Agriculture and Soil Organic Carbon: Principles, Processes, Practices and Policy Options”

Authors discussed Conservation Agriculture and Soil Organic Carbon management as adoption strategies for sustainable agriculture. It is widely accepted that Conservation Agriculture is a holistic approach to cropping system management, intended for sustainable crop production without compromising environment.

Overall manuscript is well written; however, revision is required before the formal publication of manuscript.

General Comments

Minor English revision required to improve the manuscript and for common readers. Several place word replacements are required.

Reply. We checked the MS and improved the language style as much as possible.

Define abbreviation at first appearance, subsequently use abbreviated form e.g., SOC

Reply. We checked the MS and included the meaning of abbreviations at the first appearance. We also followed the Editor suggestion about including again the meaning of abbreviation at first appearance in the concluding remarks.

Figures used in the manuscript are directly added or adopted from already published articles, clearly mention at bottom of figure.

Reply. The source was always reported in all tables and figures, and some of them are own elaborations of published data. However, for the other figures that were not elaborated from us, we considered the following options during the revision.

1) We deleted or replaced those figures that were already published elsewhere and were not really necessary in this review, leaving the reference to the sources in the main text. This is the case of former figure 6 (section 5.2 of the revised version), and figures 10 to 13 of Section 6. Policy options that were replaced with an own elaboration based on the cited literature (figure 8 in the revised version).

2) We transformed the figures into tables. This is the case of one figure in section 3.13. of the original MS (please, note that by mistake it was indicated as Figure 1), that was transformed to table 4 (section 3.3.2. of the revised version).

Incorporate below mentioned important studies, if possible

Reply. See the specific comment on Suggested citations.

Conclusion is too long

Reply. We have changed the heading of the conclusion section to better capture its contents, address the comment to incorporate future prospects and shortened the length (as requested by Reviewer 2).

Specific comments

L-40: revise as “progressive soil degradation over time”

Reply. We rephrased as suggested.

L-43: better use “soil biota”

Reply. We rephrased as suggested.

L-87-88: Revised the text

Reply. Revised as requested.

L-142-151: Add suitable reference/s

Reply. Added as suggested.

L-200: Add “)”

Reply. Added as suggested.

L-208-209: Briefly add reasons of resilience against extreme rainfall events, droughts and warming events, to facilitate common readers

Reply. We have already explained the reasons why soils increase their resilience to extreme weather events as requested for improving readers understanding (lines 625-635 in the revised version of the MS)

L-212: revise the text

Reply. We have revised the text and made some re-phrasement for improving readers understanding.

L-219: replace “hamper” with “restrict”.

Reply. Replaced as suggested.

Suggested Citations

Reply. Among the citations suggested by the Reviewer, we included the following in the Policy section as references 166 and 167.

Panagos, P.; Montanarella, L.; Barbero, M.; Schneegans, A.; Aguglia, L.; Jones, A. Soil priorities in the European Union, Geoderma Reg. 2022, 29, e00510,

Panagos, P.; Van Liedekerke, M.; Borrelli, P.; Köninger, J.; Ballabio, C.;Orgiazzi, A.; Lugato, E.; Liakos, L.; Hervas, J.; Jones, A.;  Montanarella, L. European Soil Data Centre 2.0: Soil data and knowledge in support of the EU policies. Eur. J. Soil Sci. 2022, 73, e13315.

Panagos, P., M. Van Liedekerke, P. Borrelli, J. Köninger, C. Ballabio, A. Orgiazzi, E. Lugato, L. Liakos, J. Hervas, A. Jones and L. Montanarella, 2022: European Soil Data Centre 2.0: Soil data and knowledge in support of the EU policies. European Journal of Soil Science 73, e13315.

Köninger, J., P. Panagos, A. Jones, M. J. I. Briones and A. Orgiazzi, 2022: In defence of soil biodiversity: Towards an inclusive protection in the European Union. Biological Conservation 268, 109475.

Borrelli, P., P. Panagos, C. Alewell, C. Ballabio, H. de Oliveira Fagundes, N. Haregeweyn, E. Lugato, M. Maerker, J. Poesen, M. Vanmaercke and D. A. Robinson, 2023: Policy implications of multiple concurrent soil erosion processes in European farmland. Nature Sustainability 6, 103-112.

Panagos, P., L. Montanarella, M. Barbero, A. Schneegans, L. Aguglia and A. Jones, 2022: Soil priorities in the European Union. Geoderma Regional 29, e00510.

Borrelli, P., P. Panagos, C. Alewell, C. Ballabio, H. de Oliveira Fagundes, N. Haregeweyn, E. Lugato, M. Maerker, J. Poesen, M. Vanmaercke and D. A. Robinson, 2023: Policy implications of multiple concurrent soil erosion processes in European farmland. Nature Sustainability 6, 103-112.

Panagos, P., E. Lugato, C. Ballabio, I. Biavetti, L. Montanarella and P. Borrelli, 2022: Soil Erosion in Europe: From Policy Developments to Models, Indicators and New Research Challenges. In: R. Li, T. L. Napier, S. A. El-Swaify, M. Sabir and E. Rienzi eds. Global Degradation of Soil and Water Resources: Regional Assessment and Strategies. pp. 319-333. Springer Nature Singapore, Singapore.

Round 2

Reviewer 1 Report

No more comment.

Reviewer 2 Report

As the authors have addressed all the suggested points, the manuscript is okay from my side and may be accepted.